**Data Availability Statement:** All relevant data are within the manuscript and its Supporting information files. The raw data were submitted to

# Proteome and morphological analysis show unexpected differences between promastigotes of *Leishmania amazonensis* PH8 and LV79 strains

**Fabia Tomie Tano**[1], **Gustavo Rolim Barbosa**[1], **Eloiza de Rezende**[1], **Rodolpho Ornitz Oliveira Souza**[1], **Sandra Marcia Muxel**[2], **Ariel Mariano Silber**[1], **Giuseppe Palmisano**[1], **Beatriz Simonsen Stolf**[1]*

1 Department of Parasitology, Institute of Biomedical Sciences, University of São Paulo, São Paulo, Brazil,
2 Department of Immunology, Institute of Biomedical Sciences, University of São Paulo, São Paulo, Brazil

* bstolf@usp.br

## Abstract

### Background

Leishmaniases are diseases caused by *Leishmania* protozoans that affect around 12 million people. *Leishmania* promastigotes are transmitted to vertebrates by female phlebotomine flies during their blood meal. Parasites attach to phagocytic cells, are phagocytosed and differentiate into amastigotes. We previously showed that PH8 and LV79 strains of *Leishmania amazonensis* have different virulence in mice and that their amastigotes differ in their proteomes. In this work, we compare promastigotes' infectivity in macrophages, their proteomes and morphologies.

### Methods/Principal findings

Phagocytosis assays showed that promastigotes adhesion to and phagocytosis by macrophages is higher in PH8 than LV79. To identify proteins that differ between the two strains and that may eventually contribute for these differences we used a label-free proteomic approach to compare promastigote´s membrane-enriched fractions. Proteomic analysis enabled precise discrimination of PH8 and LV79 protein profiles and the identification of several differentially abundant proteins. The proteins more abundant in LV79 promastigotes participate mainly in translation and amino acid and nucleotide metabolism, while the more abundant in PH8 are involved in carbohydrate metabolism, cytoskeleton composition and vesicle/membrane trafficking. Interestingly, although the virulence factor GP63 was more abundant in the less virulent LV79 strain, zymography suggests a higher protease activity in PH8. Enolase, which may be related to virulence, was more abundant in PH8 promastigotes. Unexpectedly, flow cytometry and morphometric analysis indicate higher abundance of metacyclics in LV79.

PRIDE (https://www.ebi.ac.uk/pride/archive/) under the submission number PXD017870.

**Funding:** This work was supported by FAPESP (BSS: 2018/14972-8; FT: 2017/26197-6; GP: 2014/06863-3, 2018/18257-1, 2018/15549-1), CNPq (GP: Productivity fellowship), CNPq grant 403100/2021-6 and CAPES. The funders had no role in study design, data collection and analysis, decision to publish, or preparation of the manuscript.

**Competing interests:** The authors have declared that no competing interests exist.

## Conclusions/Significance

Proteome comparison of PH8 and LV79 promastigotes generated a list of differential proteins, some of which may be further prospected to affect the infectivity of promastigotes. Although proteomic profile of PH8 includes more proteins characteristic of metacyclics, flow cytometry and morphometric analysis indicate a higher abundance of metacyclics in LV79 cultures. These results shed light to the gaps in our knowledge of metacyclogenesis in *L. amazonensis*, and to proteins that should be studied in the context of infection by this species.

## Introduction

Leishmaniases are diseases caused by more than 20 species of *Leishmania* from *Leishmania* and *Viannia* subgenera. According to WHO, 98 countries are endemic for leishmaniasis, which can be classified according to its clinical manifestations into tegumentary (TL) and visceral (VL) forms, endemic in 89 and 79 countries, respectively [1].

The clinical form and severity of the disease depend on the parasite species and isolate, as well as vector and mammalian host characteristics [2–4]. *Leishmania infantum* and *Leishmania donovani* are the most frequent etiologic agents of VL, the systemic and most severe form of the disease. Many species may cause TL, which can be further classified into cutaneous localized, mucosal, diffuse and disseminated forms [4, 5]. In Brazil, seven dermotropic species are associated with TL, and *Leishmania braziliensis*, *Leishmania amazonensis* and *Leishmania guyanensis* are the most frequent ones [6]. *Leishmania amazonensis* symptomatic infections usually lead to cutaneous localized leishmaniasis, characterized by one or a few ulcerated lesions, and more rarely to the diffuse form, characterized by the absence of cell-mediated immunity and typified by many non-ulcerated lesions with high parasite loads [4].

*Leishmania* parasites have a heteroxenic life cycle. They are transmitted by female phlebotomine of the *Phlebotomus* (Africa, Asia and Europe) and *Lutzomyia* (Americas) genera [7]. The insect vector ingests macrophages infected with amastigotes from the vertebrate host during its blood meal. In the vector digestive tract, amastigotes differentiate into procyclic promastigotes, which multiply in the midgut [8]. Promastigotes then migrate to the thoracic midgut and stomodeal valve, where they multiply again and start differentiation into infective metacyclic forms [8]. During another blood meal, the infected vector transmits the parasite to a new mammalian host. Metacyclic promastigotes are then internalized by several phagocytic cells and perpetuate mainly in macrophages [9]. The receptors involved in the recognition and phagocytosis of parasites are mainly complement (CR), mannose, fibronectin and Fc-γ receptors [10]. Inside the macrophage, the parasite converts into the amastigote form, which resides and proliferates in an acidic vacuole [9].

Promastigotes and amastigotes molecules having a key role in the establishment of infection in the insect vector and mammalian host are named virulence factors [11]. They include membrane proteins and glycoconjugates (Glycosylphosphatidylinositol (GPI)-anchored glycoproteins, lipophosphoglycan (LPG), free GPI glycolipids and proteophosphoglycans). An abundant molecule in promastigote´s surface is LPG, a GPI-anchored glycoconjugate composed of a long phosphoglycan chain, a glycan core and a neutral oligosaccharide cap [11]. LPG is essential for adhesion of some *Leishmania* species to some vectors [8], as well as for phagocytosis and survival inside the macrophage [12]. Another important membrane-

anchored virulence factor is metalloproteinase GP63, a protein involved in phagocytosis, resistance to complement lysis and subversion of macrophage signaling and activation [13–17].

Proteomic strategies have been employed in the identification of virulence factors of different *Leishmania* species and strains and for deciphering the mechanisms involved in the increased virulence conferred by specific factors (revised in [18]). However, most of these studies compared genetically altered parasites (usually parasite knocked out for specific genes) and/ or employed techniques such as 2-DE that do not allow the analysis of a high number of proteins. A recent study from our group compared two strains of *L. amazonensis* named PH8 and LV79 [19]. Promastigotes and amastigotes of PH8 strain caused larger lesions with higher parasite loads in murine models [19]. Amastigotes from LV79 and PH8 were isolated from footpad lesions of BALB/c mice and their soluble proteomes were compared. The abundances of all proteins and of the differential ones precisely clustered samples from the same strain [19]. Thirty-seven proteins showed different abundance between the two strains, 12 of which were increased in PH8 and 25 in LV79 amastigotes. Interestingly, superoxide dismutase, HSP70 and tryparedoxin peroxidase (TXNPx), proteins associated with augmented resistance to oxidative stress and survival of *Leishmania*, were increased in the soluble extracts of the less virulent LV79 strain [19].

We then wondered if the higher parasite loads observed for PH8 lesions resulted only from higher infectivity and survival of PH8 amastigotes or if promastigotes from this strain were also more efficiently internalized by macrophages. In the present study, we compared PH8 and LV79 early stationary phase promastigotes in terms of their internalization by murine macrophages *in vitro* and showed that PH8 are more adherent and are more phagocytosed by macrophages. To try to understand the molecular basis of these phenotypic differences we compared membrane-enriched proteomes of promastigotes of the two strains and identified several proteins with differential abundances, which will be discussed throughout the paper.

The higher phagocytosis of PH8 promastigotes may be relevant in early steps of *in vivo* infection, contributing to the higher virulence of this strain. Proteomic comparison identified several proteins, some of which may be further explored as potentially involved in *L. amazonensis* binding to macrophage and phagocytosis.

## Material and methods

### Ethics statement

Experiments with BALB/c mice were performed according to the Brazilian College of Animal Experimentation (CONEP) guidelines and with the approval of the Institutional Animal Care and Use Committee (CEUA) of the University of São Paulo (protocol number 9829290419).

### *L. amazonensis* promastigotes culture

For *L. amazonensis* promastigotes culture, medium 199 (Sigma) was supplemented with 0.005% hemin, 40 mM HEPES pH 7.4, 100 μM adenine, 4 mM sodium bicarbonate, 20 μg/mL gentamicin and 10% FBS. Promastigotes of LV79 (MPRO/BR/72/M1841, isolated from the sand fly *Lutzomyia flaviscutellata* from Pará State, Brazil) and PH8 (IFLA/BR/67/PH8, obtained from the rodent *Proechimys sp* from Pará State, Brazil) strains were obtained through cultivation of amastigotes derived from BALB/c mice lesions in medium 199 at 24˚C. Parasites were subcultured weekly to the initial density of $2 \times 10^6$ parasites/mL until eighth passage. For all experiments, parasites at fourth day of culture, early stationary phase, were employed.

## Flow cytometry analysis to determine metacyclic promastigotes

$2 \times 10^6$ promastigotes were collected in each of the 5 tubes prepared for each strain, washed twice with PBS pH 7.2 (800 x g, 10 min at 4°C) and fixed in 1% paraformaldehyde in PBS for 30 min at 4°C. Next, the parasites were washed with PBS and resuspended in 50 uL PBS. Cell acquisition was performed using a BD LSR Fortessa Cell analyser (BD, Franklin Lakes, NJ, USA), and the collected data were analyzed using FlowJo Software (LLC, Ashland, OR, USA). The frequencies of metacyclic promastigote forms were determined by gating cells with morphology similar to previously described [20] from at least 25,000 events, based in forward scatter (FSC, voltage 600) and side scatter (SSC, voltage 350) features.

## Morphometric analysis of promastigotes

Morphometric analysis of PH8 and LV79 promastigotes was performed by collecting parasites during *in vitro* culture. Cultures were synchronized by sub culturing 3 times for $2 \times 10^6$/mL every 3 days, and parasites were collected at days 2, 4 and 6 of culture, counted, fixed and stained with Giemsa. 150 parasites per condition were classified in procyclics, nectomonads, leptomonads or metacyclics based on body length and flagellum length, according to [21].

## Phagocytosis assay

BALB/c mice were euthanized in a $CO_2$ chamber and cleaned with 70% alcohol. With the aid of tweezers, scalpel and scissors, skin and muscles of the hindlimb were removed in order to obtain its tibia and femur. After hygiene in 70% alcohol, the bones were washed in sterile PBS and epiphyses were excised exposing their medullary cavity. With the help of a 21-gauge needle connected to a 10 mL syringe, the bone marrow cells were expelled from the cavity by the passage of RPMI 1640 medium. The cells were homogenized in RPMI 1640 medium supplemented with 20% FBS, 30% L929 supernatant and 20 µg/mL gentamicin, distributed in 75 cm² culture flasks and incubated at 37°C and 5% $CO_2$ for seven days for differentiation into macrophages.

Cells were harvested from culture flasks in PBS with the aid of a cell scraper and centrifuged at 800 x g for 10 minutes. The resulting pellet was resuspended in RPMI 1640 medium supplemented with 10% FBS and 20 µg/mL gentamicin. Cells stained with 0.4% trypan blue were counted using a hemocytometer and $4 \times 10^5$ viable cells were plated in 13 mm diameter glass coverslips arranged in a 24-well plate. After overnight incubation at 37°C 5% $CO_2$, macrophages were incubated with parasites at multiplicity of infection (MOI) of 10:1 in RPMI 1640 medium supplemented with 10% FBS and 20 µg/mL gentamicin at 4°C for 2 hours. Subsequently, the plates were incubated at 34°C 5% $CO_2$ for 5 minutes and again transferred to ice. Cells were washed with ice cold PBS and fixed with PBS containing 4% paraformaldehyde for 5 minutes. After washing with PBS, cells were incubated with 50 mM ammonium chloride for 30 minutes and blocked with PBS 1% BSA (Gibco, Thermo Fisher Scientific) for 30 minutes at 37°C.

Cells were incubated overnight with mice anti-*Leishmania* serum (1:75, "homemade") before permeabilization, therefore antibodies bind exclusively to adhered parasites. In the following day, cells were washed with PBS and permeabilized with 0.1% Triton X-100 in TBS 1% BSA for 10 minutes, washed with PBS and then incubated with a mixture of 10 mg/mL DAPI (1:600), anti-mouse IgG (H+L) secondary antibody Alexa Fluor®488 (1:1,000; Molecular Probes, Thermo Fisher Scientific) and Texas Red 568 phalloidin (1:500; Molecular probes, Thermo Fisher Scientific) for 1 hour at room temperature. Slides were washed six times with PBS, three times with water and mounted with Prolong diamond (Molecular Probes, Thermo Fisher Scientific) for fluorescence analysis. Images were acquired in a DMI6000B/AF6000

(Leica) fluorescence microscope coupled to a digital camera system (DFC 365 FX). Tests were performed in technical triplicate and a total of 500 macrophages were analyzed for each coverslip. Parasites were classified as adhered (green with nucleus in blue) and phagocytosed (nucleus labeled in blue).

## Protein extraction

Membrane-enriched fractions were obtained through sodium carbonate extraction adapted from [22–25]. Briefly, 1 x $10^9$ promastigotes were washed three times with 4 mL of PBS and resuspended to a final density of 4 x $10^9$ promastigotes/mL in a 100 mM sodium carbonate (pH 11) solution with protease inhibitor cocktail (800 nM aprotinin, 50 μM bestatin, 1 mM AEBSF, 15 μM E64, 20 μM leupeptin e 10 μM pepstatin A; Fermentas, Thermo Fischer Scientific). Promastigotes were lysed by eight cycles of freeze and thaw (liquid nitrogen and 40˚C) and ultrasonication, performed three times for five seconds at 40% output spaced with 30-second intervals on ice (Vibra Cell VC50, Sonics&Materials Inc.). Remaining whole parasites were removed by centrifugation at 2,000 x g for 5 minutes at 4˚C and the resulting supernatant was ultracentrifuged at 120,000 x g for 1 hour at 4˚C. Lastly, supernatant (cytoplasmic fraction) was transferred to a new microtube and pellet (membrane-enriched fraction) was resuspended in 8 M urea and 100 mM ammonium bicarbonate (pH 7.5) with protease inhibitor cocktail (Fermentas). Protein concentration was determined by Bradford assay (BioRad).

For total protein extracts (used in SDS-PAGE, western blot and zymography), promastigotes were resuspended in PBS with protease inhibitor cocktail (Fermentas) at a final density of 2 x $10^9$ parasites/mL. Parasites were lysed by eight cycles of freeze and thaw (liquid nitrogen and 40˚C). Protein concentration was determined by Bradford assay.

## SDS-PAGE, western blot and zymography

15 or 20 μg of protein were added to sample buffer (2% SDS; 60 mM Tris-HCl pH 6.8; 0.1% bromophenol blue; 1.2% β-mercaptoethanol; 10% glycerol) and boiled at 95˚C for 5 minutes prior to separation in 10% or 12% acrylamide gels. Gels were composed of a stacking gel (5% acrylamide/bis-acrylamide; 125 M Tris-HCl pH 6.8; 0.1% SDS; 0.1% ammonium persulfate; 0.1% TEMED) and a running gel (375 mM Tris HCl pH 8.8; 10 or 12% acrylamide/bis-acrylamide; 0.1% SDS; 0.1% ammonium persulfate; TEMED 0.04%). Electrophoresis was performed in running buffer (25 mM Tris-HCl; 250 mM glycine; 0.1% SDS) at 80 V for 30 minutes and at 120 V for the rest of the run. For staining, gels were incubated with staining solution (40% methanol, 10% acetic acid and 0.1% Coomassie Brilliant blue R-250) for one hour at room temperature and then washed with a 20% methanol and 5% acetic acid solution for destaining.

For western blot, proteins were transferred to nitrocellulose membranes (GE healthcare) in transfer buffer (25 mM Tris pH 8.2; 192 mM glycine; 20% methanol; 0.1% SDS) employing the TE 77 semidry system (GE healthcare). In order to prevent nonspecific binding, membranes were blocked with PBS with 5% milk and 0.1% Tween 20 for 1 hour at room temperature prior to incubation overnight at 4˚C with primary antibodies (anti-GP63 (1: 5,000), anti-TXNPx (1: 5,000), anti-enolase (1: 250) or anti-α-Tubulin (1: 50,000; T5168 Sigma)) diluted in PBS with 2.5% milk and 0.1% Tween 20. After three washing steps with PBS 0.1% Tween 20, membranes were incubated for 1 hour at room temperature with anti-mouse IgG (1:10,000; KPL) HRP-conjugated secondary antibodies diluted in PBS with 2.5% milk and 0.1% Tween 20. Membranes were washed three times with PBS 0.1% Tween 20 and two times with PBS. Lastly, membranes were incubated with ECL Prime Western Blotting Detection Reagent (GE healthcare) and chemiluminescence was detected by the ChemiDoc XRS+ Imaging system (BioRad).

Band intensities were determined through ImageJ software. For validation, western blots were performed with biological triplicates.

For zymography, 2 μg of protein in sample buffer without β-mercaptoethanol were loaded to 12% SDS gels containing 0.1% gelatin. After electrophoresis, gels were incubated for 1 hour at room temperature in buffer containing 50 mM Tris pH 7.4, 2.5% Triton X-100, 5 mM CaCl$_2$ and 1 μM ZnCl$_2$. A parallel experiment was performed without ZnCl$_2$. Subsequently, gels were incubated overnight at 37˚C in buffer containing 50 mM Tris pH 7.4, 5 mM CaCl$_2$, 1 μM ZnCl$_2$ and 0.01% NaN$_3$ [26]. Gels were stained and destained as described above. Zymography was performed with biological triplicates.

## Protein digestion and peptide desalting

Tryptic digestion, desalting and MS were performed according to [27]. Briefly, after adjusting the pH of the samples to 7.5 with ammonium bicarbonate, 100 μg of protein were reduced by incubation with 10 mM DTT at 30˚C for 30 minutes. Subsequently, proteins were alkylated by incubation with 40 mM iodoacetamide for 30 minutes in the dark at room temperature. To stop the alkylation reaction, DTT was added to a final concentration of 10 mM. Digestion buffer (50 mM ammonium bicarbonate and 10% ACN) was added to dilute urea concentration to 1.4 M. Digestion was performed by incubation with trypsin 1:50 (w/w) for 16 hours at 30˚C and stopped by the addition of 10% formic acid to a final concentration of 1%.

The resulting peptide mixtures were desalted with hydrophilic–lipophilic-balanced SPE (Waters) and peptides eluted with 1 mL of 70% (v/v) ACN and 1% (v/v) TFA. Finally, the peptides were vacuum dried before MS analysis.

## MS analysis

Promastigote membrane proteome analysis was performed with three biological replicates for each strain. Each biological replicate was analyzed twice by nLC-MS/MS, resulting in two technical replicates. Peptide separation was performed on the EASY- nLC system (Thermo Fisher Scientific) employing the Acclaim PepMap 100 C18 column (10 cm; 75 μm ID; 3 μm C18-A2; Thermo Fisher Scientific) and solvents A (0.1% formic acid) and B (0.1% formic acid; 99% ACN). Elution was performed for 77 minutes using a linear gradient from 1 to 50% of solvent B with a flow rate of 0.3 μL/min. Columns were washed and re-equilibrated between experiments. The eluted peptides were ionized by nanoelectrospray and directed to the LTQ Velos Orbitrap mass spectrometer (Thermo Fisher Scientific). Mass spectra were acquired in positive-ion mode with a range of 400 to 1,600 m/z, resolution of 30,000 (full width at half-maximum at $m/z$ 400) and AGC target $>1 \times e^6$. The 20 most abundant precursor ions ($z \geq 2$) were isolated to a target value of 5,000 and isolation width of 2 and fragmented by low-energy CID (normalized collision energy of 35% with 10 ms activation time) in the linear IT. For the detection of less abundant peptides, dynamic exclusion was applied with exclusion list size of 500, exclusion duration of 30 s and repeat count of 1. The raw data were submitted to PRIDE (https://www.ebi.ac.uk/pride/archive) under the submission number PXD017870.

## Protein identification and bioinformatics analyses

Protein identification and bioinformatics analysis were performed with the MaxQuant software version 1.5.3.8 [28–30]. RAW files of the technical duplicates were imported to the software and combined into one. Search engine Andromeda was used to search MS/MS spectra against Uniprot *Leishmania mexicana* database (2018/09/10, 8,044 entries). For the search, the following parameters were adopted: (i) 4.5 ppm tolerance level for MS and 0.5 Da for MS/MS; (ii) trypsin cleavage at both ends with maximum of two missed cleavages allowed per peptide;

(iii) cysteine carbamidomethylation (57.021 Da) as a fixed modification; (iv) oxidation of methionine (15.994 Da) and protein N-terminal acetylation (42.010 Da) as variable modifications. The feature 'match between runs' was employed with a 0.7 minute match time and 20 minute alignment time window. All identifications were filtered to obtain a peptide and protein false discovery rate (FDR) of less than 1%. For protein identification, at least one unique peptide was required. For label-free quantification using unique and razor peptides, a minimum of two ratio counts was established as necessary. Protein abundance was calculated based on its normalized spectral intensity (LFQ Intensity).

Statistical analyses were performed with the Perseus software v.1.5.8.5 [31]. Potential contaminants, reverse hits and proteins identified only by the site were removed. The LFQ Intensity values were transformed into log2 (X) and only proteins with at least two valid values in at least one of the groups (PH8 or LV79) were maintained. Imputation was performed separately for each expression column; missing values were replaced by random numbers drawn from a normal distribution adopting a down shift of 1.8 and a distribution width of 0.3 [31]. Principal component analysis was constructed with all proteins identified. After exclusion of exclusively detected proteins, statistical analyses to compare LV79 and PH8 strains were performed by Student's *t* test with Benjamini-Hochberg correction and FDR = 0.05. Differences were considered significant for corrected *p*-value (*q*-value) $\leq$.05. The heat map was constructed based on the hierarchical clustering of the Z-scores calculated from the log2 of LFQ Intensity values of differentially abundant proteins. For fold change calculations, mean of LFQ Intensity values of PH8 biological replicates were divided by the mean of LFQ Intensity values of LV79 biological replicates. Biological processes in which proteins participate were determined according to Gene ontology and KEGG Enzyme annotations and other literature sources. Protein subcellular localization was determined according to gene ontology (Cellular Component) and prediction tools classifications (DeepLoc-1.0 (http://www.cbs.dtu.dk/services/DeepLoc/), WoLF PSORT (https://wolfpsort.hgc.jp/), LocTree3 (https://rostlab.org/services/loctree3)) and TOP-CONS (http://topcons.cbr.su.se/pred/)) [32–35].

## Enolase activity assay

For enzymatic assays, parasites were washed two times with PBS and resuspended in lysis buffer (20 mM Tris HCl pH 7.7; 1 mM EDTA; 0.25 M sucrose; 0.1% Triton X-100) with protease inhibitor cocktail (Fermentas) to a final density of $2.5 \times 10^9$ parasites/mL. Samples were homogenized in vortex and centrifuged at 15,000 x *g* for 15 minutes at 4°C. Supernatant's protein content was measured at 280 nm by Nanodrop. Reactions were initiated by addition of 40 μg of protein to the mixture containing 50 mM Tris–HCl pH 7.5, 2 mM $MgSO_4$, 75 mM KCl and 2 mM phosphoenolpyruvate (PEP) in a final volume of 200 μL in 96-well plates. This assay monitors the enolase reverse reaction, following PEP conversion to 2-phosphoglycerate (2-PGA), measured spectrophotometrically at 240 nm in the SpectraMax i3 multi-mode detection platform (Molecular devices). Absorbance was converted to μmols of $PEP_{ox}$/mg.min using the molar extinction coefficient of PEP = 1.256. Enzymatic assay was performed with five biological replicates.

## Real time RT-PCR

Promastigotes ($5 \times 10^7$) from three synchronized cultures from each strain were collected at days 2 and 4 and RNA was prepared using TRIzol reagent (Life Technologies, Carlsbad, CA, USA), according to the manufacturer's instructions. RNA concentration was determined by spectrophotometry (Nanodrop ND1000, Thermo Fisher Scientific). Samples with 2,5μg were

treated with DNase I (Thermo Fisher Scientific) and used to prepare cDNA using Supescript II enzyme (Thermo Fisher Scientific) and oligodT and random hexamers.

cDNA was used as a template for qPCR with the Maxima SYBR Green/ROX qPCR Master-Mix (Thermo Fisher Scientific) and specific primers for selected CDSs (SHERP For 5' AAGGGACCAGATGAGCAACGT 3' and Rev 5' TTCAATCGTGTTGCCCACTGC 3', META 1 For 5' AAGCTTGATTGGCAAGCACAG 3' and Rev 5' CGTTCATGAAGTTCGCCACTT 3', GAPDH For 5' TCAAGGTCGGTATCAACGGC 3' and Rev 5' TGCACCGTGTCG-TACTTCAT 3'). qPCR reactions were performed on a StepOne Plus thermocycler (Thermo Fisher Scientific) using the program: 95∘C for 10 min followed by 40 cycles at 95∘C for 15 s, 60∘C for 60 s, and 72∘C for 20 s. After determining the efficiency of amplification of the four genes, 2−ΔΔCt equation was employed to calculate the relative expression [36] of each gene in all samples using GAPDH as reference gene. LV79 day 2 mean values were used for normalization. Three biological replicates of each condition and three technical replicates for each sample were analyzed.

## Statistical analysis

Statistical analyses were performed by unpaired parametric Student's *t* test (n ≤ 2) when two samples were compared, and by ANOVA followed by Tukey posttest for three or more samples. Differences were considered significant for *p*-value ≤ 0.05.

## Results

### Phagocytosis assay

A previous study of our group showed that *L. amazonensis* amastigotes of PH8 strain were more infective in mice than those of LV79 strain [19]. Moreover, proteome analysis indicated that lesion-derived amastigotes from the two strains differ in terms of the abundance of several proteins [19]. We then wondered whether promastigotes of the two strains would also differ in terms of internalization by macrophages. In face of the low proportion of metacyclics in stationary phase (around 5% according to [37]), the isolation of membrane fractions for proteomic analysis from metacyclics would not be feasible. For this reason, all experiments employed promastigotes at the fourth day of culture, which corresponds to the early stationary phase for both strains, as shown in S1 Fig. This figure also indicates that PH8 and LV79 have similar growth curves, suggesting that their populations possibly have similar compositions.

The results shown in Fig 1 indicate that PH8 promastigotes adhere more (Fig 1A) and are more efficiently phagocytosed by murine bone marrow-derived macrophages *in vitro* (Fig 1B), as shown by the higher proportion of parasites attached to the cells and internalized.

### Validation of sodium carbonate extraction for membrane enrichment

To identify membrane proteins that might account for the higher adhesion and phagocytosis of PH8 strain, we adapted a membrane extraction protocol previously employed to enrich *L. donovani* promastigotes´ membranes [22–25]. Total, cytoplasmic and membrane-enriched fractions of LV79 promastigotes were analyzed by SDS-PAGE, and a clear difference was observed in the protein profile of membrane-enriched fraction (Fig 2A). Membrane enrichment was validated by comparing the abundance of GP63, usually associated with promastigotes´ membranes, and TXNPx, a cytoplasmic enzyme, in the three fractions (Fig 2B). As expected, GP63 was more abundant in membrane-enriched when compared to the cytoplasmic fraction, while TXNPx was more abundant in the cytoplasm (Fig 2B), indicating that the extraction protocol was successfully adapted to *L. amazonensis* promastigotes.

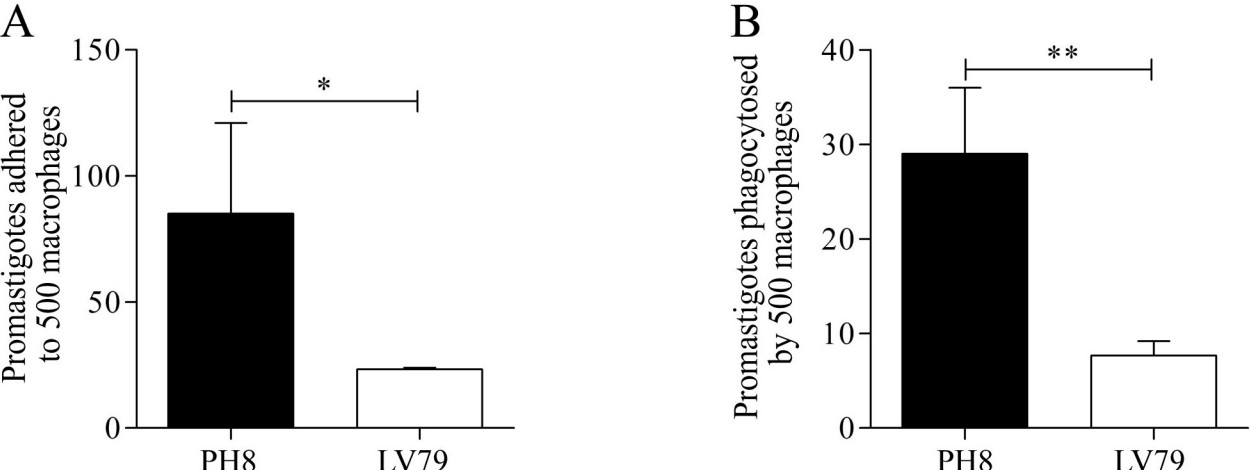

**Fig 1. *In vitro* phagocytosis assay with PH8 and LV79 promastigotes.** A. Number of promastigotes of PH8 and LV79 strains adhered to 500 macrophages. B. Number of promastigotes of PH8 and LV79 strains phagocytosed by 500 macrophages. Phagocytosis assays were performed with murine bone marrow-derived macrophages and a MOI of 10:1. Data represent mean ± SD of three technical replicates. Statistical analysis by Student's *t*-test, *: $p < .05$, **: $p < .01$, ***: $p < .001$. Representative results of two experiments with similar profile.

## Quantitative large-scale comparison of membrane-enriched proteomes of PH8 and LV79 promastigotes

Membrane-enriched fractions were prepared in biological triplicate from paired PH8 and LV79 promastigote day four cultures. Membrane enrichment was confirmed by Western blot with anti-GP63 and anti-TXNPx antibodies before proteomic analysis, as shown in S2 Fig.

Proteomes of the three samples of each strain were analyzed and 1659 proteins were identified, listed in S1 Table (Fig 3A). Principal component analysis based on all identified proteins efficiently clustered PH8 and LV79 samples (Fig 3B). From these, 1557 were identified

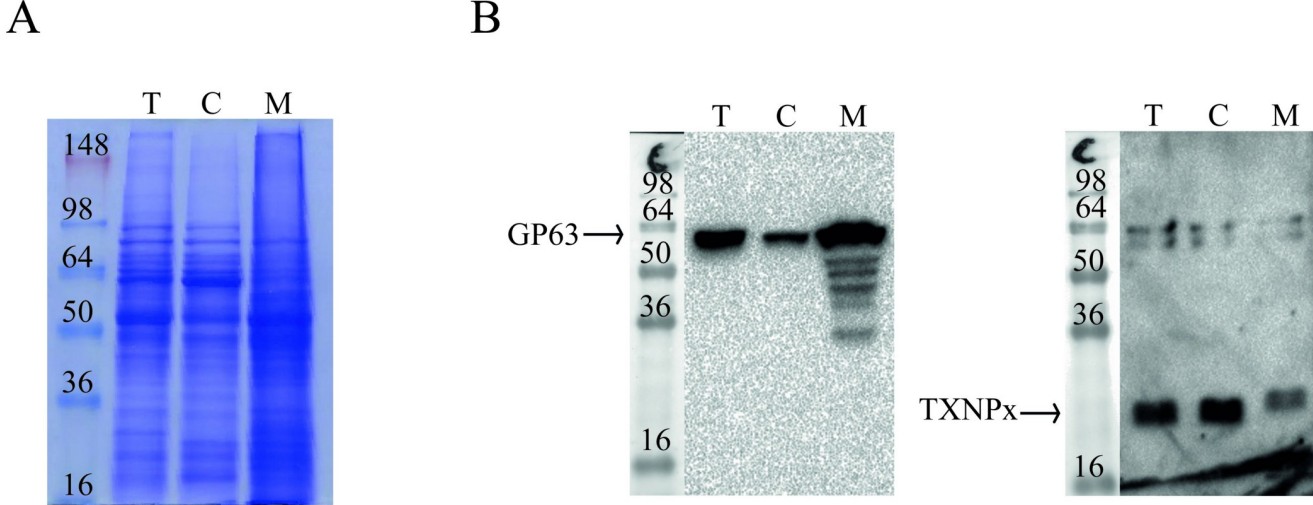

**Fig 2. Validation of sodium carbonate extraction with LV79 promastigotes.** A. 20 μg of protein of total cell (T), cytoplasmic (C) and membrane-enriched (M) fractions were analyzed by SDS-PAGE. B. Abundance of GP63 and TXNPx in 15 μg of total (T), cytoplasmic (C) and membrane-enriched (M) fractions were compared by Western blot.

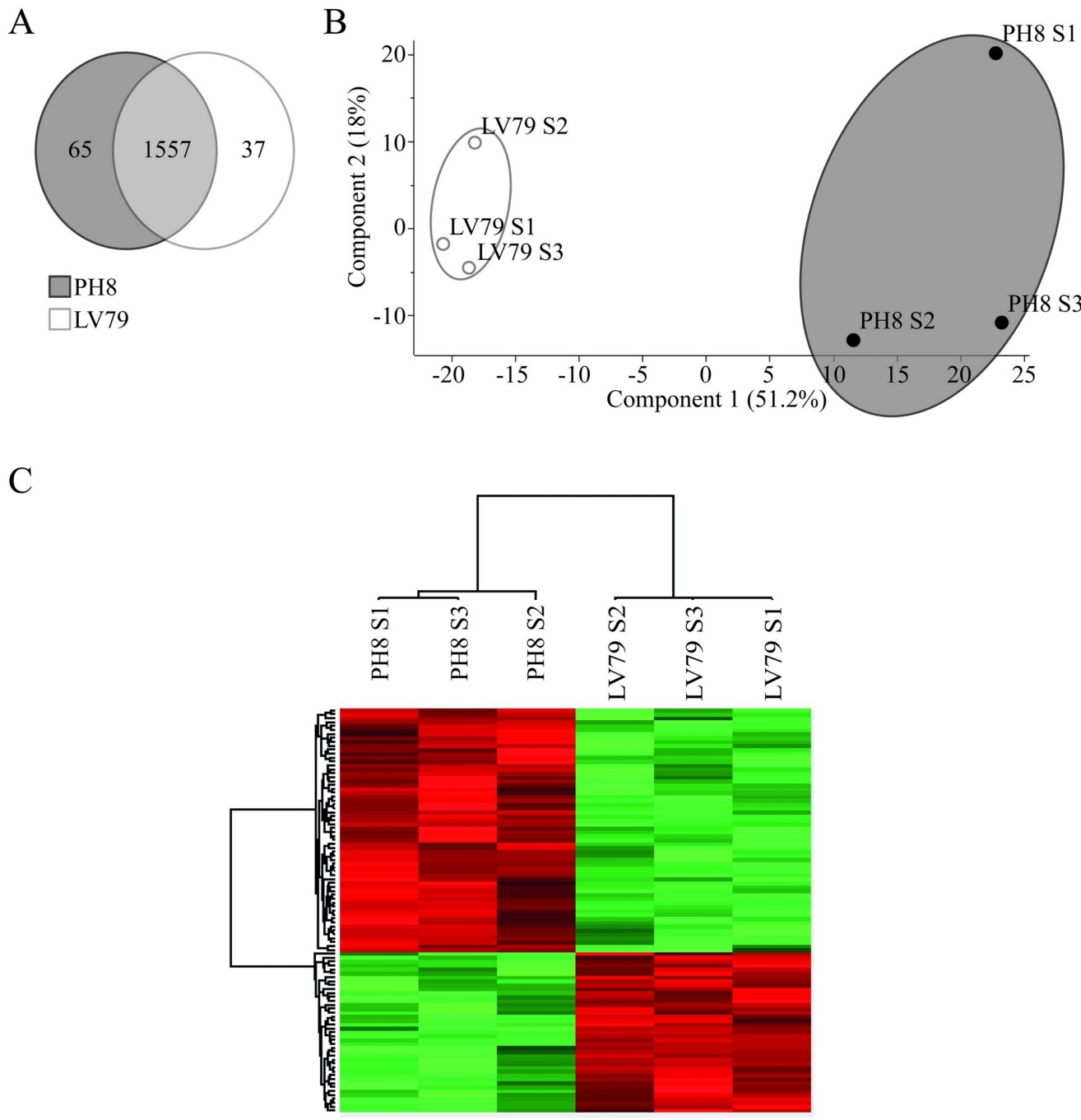

**Fig 3. Comparison of membrane-enriched proteomes of PH8 and LV79 promastigotes.** A. Venn diagram showing the number of proteins identified exclusively in PH8, in LV79 and identified in both strains. B. Clustering of the six samples (three biological samples (S1, S2 and S3) for each strain) by principal component analysis of all proteins identified. C. Heat map constructed based on the hierarchical clustering of the 6 samples (three samples (S1, S2 and S3) for each strain) based on Z-scores calculated from the log2 of LFQ Intensity values of differentially abundant proteins identified after Student's *t* test with Benjamini-Hochberg correction and FDR = 0.05.

both in PH8 and LV79 fractions, while 65 were detected exclusively in PH8 and 37 in LV79 (Fig 3A). Proteins detected exclusively in PH8 or LV79 are listed in S2 and S3 Tables, respectively. It should be noted that proteins detected exclusively in one condition might be present at levels below the limit of detection in the other condition. The majority of these proteins (33 from 65 proteins, 50.7% in PH8, and 17 from 37, 46% in LV79) has not been characterized.

Considering the proteins identified in both strains, 62 proteins were identified as more abundant in PH8 samples, while 41 were identified as more abundant in LV79, as shown in the heat map (Fig 3C). S4 Table lists the differentially abundant proteins. Four proteins showed fold increases above 10 in PH8: Putative ATP-binding cassette protein subfamily G, member 1 (LABCG1), Glutamate dehydrogenase, Putative flagellar calcium-binding protein and Surface antigen-like protein. These proteins may eventually contribute to the increased infectivity of PH8.

## Classification of proteins identified in proteomic analysis based on subcellular localization

Subcellular localization of the proteins identified in proteomic analysis was established according to Gene ontology annotations (Cellular Component). However, several *Leishmania* proteins are not characterized. Indeed, only 588 proteins from the 1659 (35,4%) had GO classification, as shown in S5 Table. In these cases, subcellular location was determined based on a consensus of DeepLoc-1.0, WoLF PSORT and Loc Tree3 predictions [32, 34, 35]. The proteins were classified as secreted or belonging to the "cilium", cytosol, golgi complex, glycosome, plasma membrane, mitochondria, nucleus, endoplasmic reticulum or intracellular vesicles (Fig 4A). In addition, predictions tools DeepLoc-1.0 and TOPCONS were utilized to identify integral membrane proteins [32, 33].

Most proteins were localized in the cytosol (43%), mitochondrion (17%) and nucleus (15%) (Fig 4A), and only 3% of the proteins were localized in the cell membrane. According to our classification, proteins that have at least one transmembrane domain correspond to 14% of the proteome (Fig 4B), most of which belong to the mitochondria (27%), cell membrane (21%) and endoplasmic reticulum (18%) (Fig 4C).

## Classification of regulated proteins based on biological process

The proteins more abundant in PH8 and LV79 promastigotes, including all exclusively detected and differentially abundant proteins, were classified according to the biological processes in which they participate. Proteins were assigned based on Gene ontology and KEGG Enzyme annotations and other literature sources into eleven biological processes: amino acid metabolism, carbohydrate metabolism, lipid metabolism, nucleotide metabolism, DNA replication and repair, transcription, mRNA processing, translation, cytoskeleton composition, membrane and vesicle trafficking and proteolysis (S2–S4 Tables; Fig 5).

We observed that proteins more abundant in PH8 promastigotes participate mainly in carbohydrate metabolism (21%), cytoskeleton composition (17%) and vesicle and membrane trafficking (13%) (Fig 5A and 5C). Proteins related to proteolysis, mRNA processing, DNA replication and repair, transcription and lipid metabolism are also more abundant in PH8 promastigotes (Fig 5C). In contrast, the majority of the proteins more abundant in LV79 promastigotes participate in translation (47%), amino acid metabolism (21%) and nucleotide metabolism (14%) (Fig 5B and 5C).

A

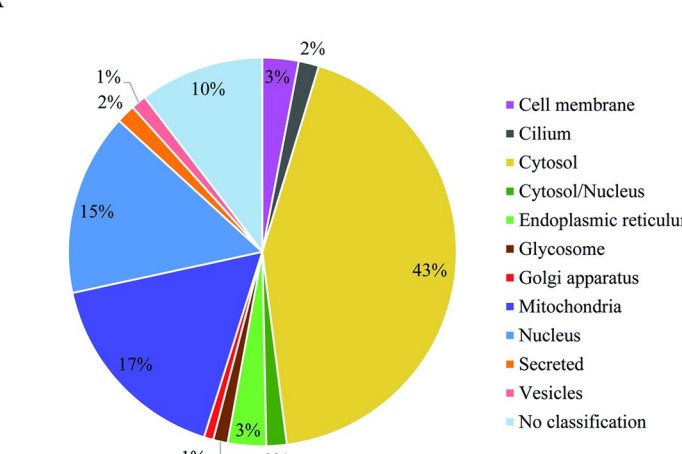

B

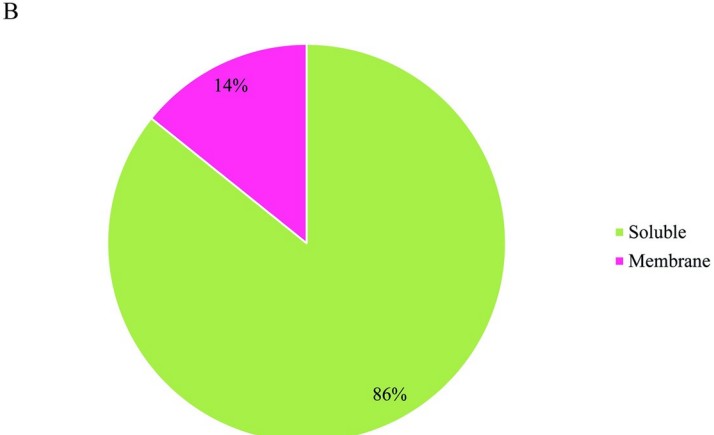

C

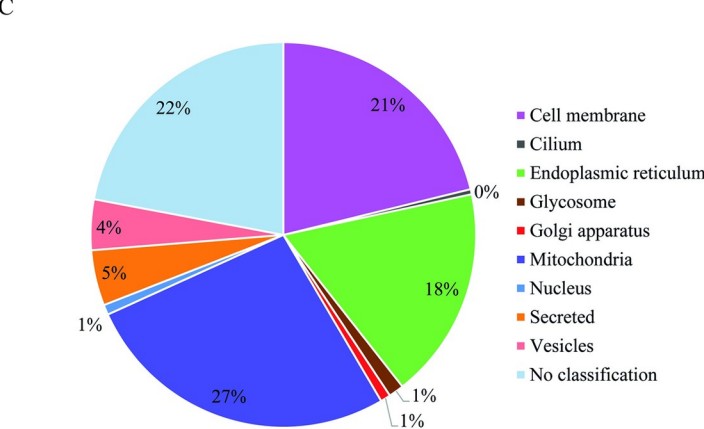

**Fig 4. Classification according to subcellular localization of all proteins identified in the membrane-enriched proteome of PH8 and LV79 promastigotes.** A. Circle chart showing percentage of proteins identified in proteomic analysis belonging to each subcellular localization. B. Circle chart showing percentage soluble and integral membrane proteins C. Circle chart showing percentage of integral membrane proteins identified in proteomic analysis belonging to each subcellular localization.

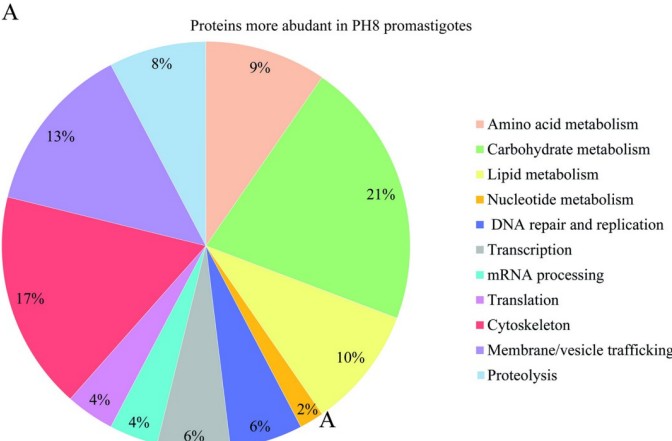

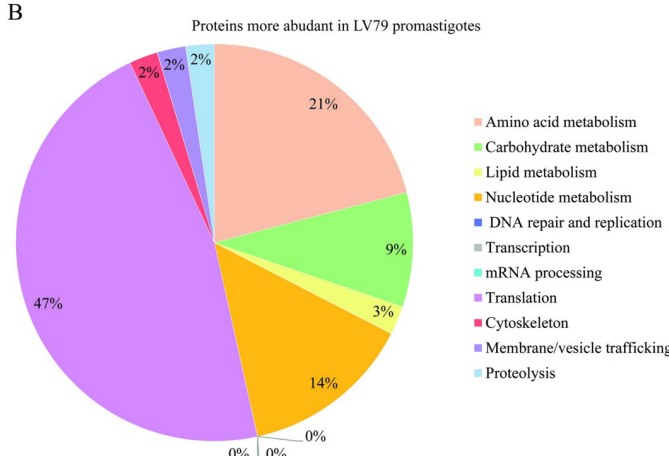

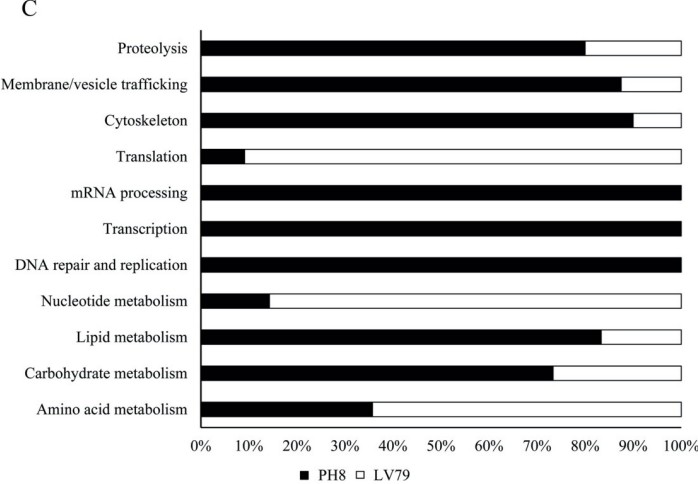

**Fig 5. Classification of differentially abundant proteins based on biological process.** A. Circle chart showing percentage of the proteins more abundant in PH8 promastigotes belonging to each biological process. B. Circle chart showing percentage of the proteins more abundant in LV79 promastigotes belonging to each biological process. C. Bar diagram showing the percentage of differential proteins from each biological process detected as more abundant in LV79 and in PH8 strains.

## GP63 is regulated at protein and enzymatic level in *L. amazonensis* PH8 and LV79 strains

Three isoforms (E9AN53, E9AN57 and E9AZL8) of GP63, a well-known virulence factor of *Leishmania*, were detected in all samples analyzed. One of these isoforms was more abundant in LV79 membrane-enriched promastigotes fractions (S4 Table and Fig 6A). This finding was not expected considering the lower virulence of this strain and the higher abundance of GP63 in soluble proteome of PH8 lesion derived amastigotes previously reported [19]. We thus compared GP63 abundance in total and membrane-enriched fractions of PH8 and LV79 promastigotes by Western blot. Fig 6B shows that GP63 is indeed more abundant in total (left) and membrane-enriched (right) fractions of LV79 when compared to PH8. We then sought to analyze GP63 activity of promastigotes total extracts (with inhibitors to several proteases other than metalloproteases) by zymography in gelatin gels. Data shown in Fig 6C (left panel), representative of three experiments with similar results, suggests that GP63 (50 kDa band) is enzymatically more active in PH8. Non-reducing Western blot for GP63 (Fig 6C, right panel) supports the size of band pointed as GP63. We have performed an assay without $ZnCl_2$ and the bands near 50 kDa were a little less pronounced, but still evident. We believe $ZnCl_2$ remnants present in total protein extract may be enough for GP63 activity, as observed by others [38, 39].

The higher abundance of enolase in the proteome of PH8 membrane-fractions is shown in S4 Table and Fig 7A. This result was validated in total (left) and membrane-enriched (right) fractions by western blot (Fig 7B). Data shown in Fig 7C, representative of three experiments with similar results, suggests that there are no significant differences between strains in terms of enolase activity, although values for PH8 tend to be higher than those of LV79 (Fig 7C).

## Can population composition differences explain higher infectivity and proteome profile of PH8?

GO annotation analysis showed that the majority of the proteins upregulated in LV79 promastigotes participate in translation, while most of the proteins upregulated in PH8 are involved in carbohydrate metabolism, cytoskeleton composition and vesicle and membrane trafficking.

To evaluate whether a different composition in terms of parasite stages in day 4 cultures could account for the discrepant proteomic profiles and infectivity observed for PH8 and LV79, we compared parasites from the two strains by flow cytometry and morphometrical analysis. The frequency of metacyclic forms in three synchronized cultures from each strain was estimated using flow cytometry. The cells gated in the region with lower Forward side scatter (FSCl$^{ow}$) and SSC features were representative of *Leishmania* (Fig 8A), as described previously [20]. Results shown in Fig 8B indicate that cultures (day 4) of LV79 strain present a significantly higher frequency of metacyclic forms (54,86%) compared to those of PH8 strain (22,37%).

Morphometric analysis of PH8 and LV79 promastigotes was performed during *in vitro* culture and parasites were classified in procyclics, nectomonads, leptomonads or metacyclics, according to [21]. Data from day 6 was not considered valid due to the huge proportion of parasites without nucleus, without flagellum or with atypical morphology. The proportion of the four stages in each culture is shown in Fig 9. Representative images of LV79 and PH8 cultures in days 2 and 4 and 6 are available as S3 Fig.

As can be observed, both cultures show a reduction of procyclics and nectomonads and an increase in leptomonads and metacyclics from log (day 2) to early stationary phase (day 4) (Fig 9), in agreement to what is observed during development in the sandfly. Besides, it is also evident that LV79 cultures at days 2 and 4 show higher proportion of metacyclics compared to

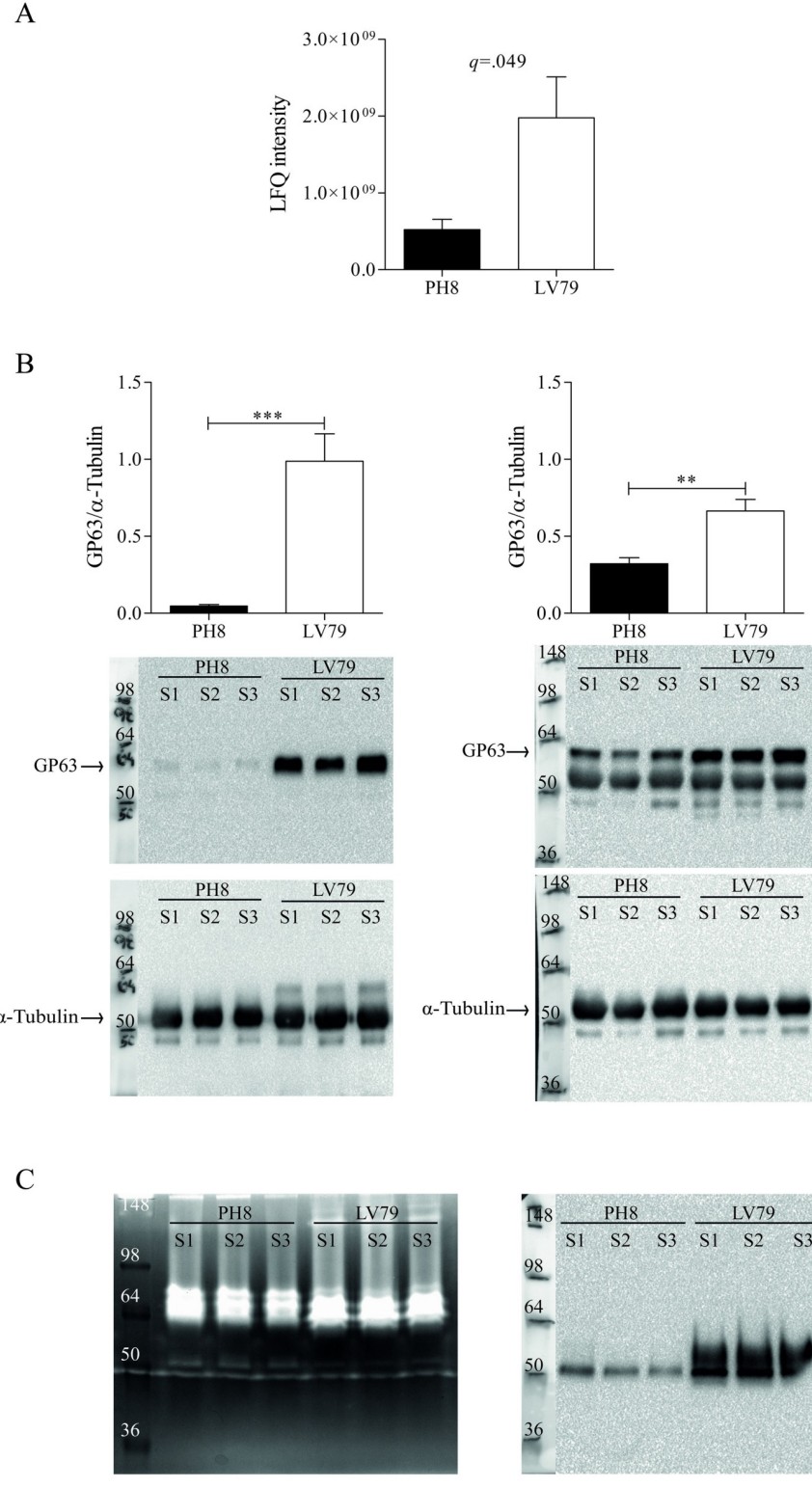

**Fig 6. GP63 abundance and activity in *L. amazonensis* promastigotes of LV79 and PH8 strains.** A. GP63 abundance (non-log transformed LFQ intensities) in proteomes of membrane-enriched fractions of LV79 and PH8 promastigotes. B. GP63 (upper figure) and α- tubulin (lower figure) abundances in total extracts (left) and membrane-enriched fractions (right) (three biological samples for each strain) of LV79 and PH8 by Western blot. For membrane-enriched fractions, GP63 is shown together with tubulin labeling in the upper figure. Graphs show normalized values

(GP63/ α- tubulin). Graph C. GP63 proteolytic activity in total extracts (three biological samples for each strain) was measured by zymography (one experiment representative of three). A non-reducing Western blot for GP63 is shown in the right. Data in A and B represent means and SD of three biological samples (S1, S2 and S3) for each strain. For A, statistical analysis was performed by Student's *t* test with Benjamini-Hochberg correction (FDR = 0.05) and the resulting *q*-value is shown in graph. For B, statistical analysis was performed by Student's *t*-test, *: $p < .05$, **: $p < .01$, ***: $p < .001$.

PH8. In fact, day 2 cultures of LV79 display 8.12% metacyclics compared to 0% for PH8, and day 4 cultures of LV79 show 16.56% of metacyclics compared to 3.07% for PH8. The higher proportion of metacyclics in day 4 cultures of LV79 agrees with data from flow cytometry (Fig 8B).

We also evaluated the abundance of transcripts of two genes commonly used as markers for metacyclogenesis in *Leishmania*: SHERP and META1. SHERP transcripts were described as good markers for *L. infantum* metacyclics isolated from sandfly [40] and are also more abundant as transcripts and proteins in late stationary cultures of *L. major* [21, 41]. META 1 protein was shown to be more abundant in *L. amazonensis* stationary cultures, and its overexpression increased parasite virulence [42].

The transcripts of these three genes were compared in day 2 and day 4 cultures of LV79 and PH8. The results of three independent experiments are plotted in Fig 10.

Despite a visible increase in SHERP and META1 transcripts in day 4 compared to day 2 cultures for both LV79 and PH8, no statistical difference was observed. Besides, there was no difference in expression of these genes between LV79 and PH8 cultures.

## Discussion

In this study, we showed that PH8 stationary phase promastigotes are more adherent to and phagocytosed by bone marrow-derived murine macrophages *in vitro* than LV79. These results parallel the higher infectivity observed for PH8 in mice *in vivo* after inoculations of stationary phase promastigotes or amastigotes [19]. We hypothesized that this phenotypic characteristic correlates with differences in membrane composition between the two strains. In order to identify proteins possibly involved in the difference of phagocytosis and adhesion, we performed a comparative high throughput proteomic analysis of membrane-enriched fractions of PH8 and LV79 promastigotes.

Despite observing membrane enrichment by comparing GP63 and TXNPx abundance in different cellular fractions, a low percentage of membrane proteins was observed. A high number of contaminating cytoplasmic proteins was expected, as it is commonly reported by studies that employ similar protocols for membrane enrichment [25, 43–45].

Proteins potentially associated with virulence such as enolase and putative ABC transporter (ATP-binding cassette) member 1 from subfamily G (LABCG1) were upregulated in PH8 promastigotes. LABCG1, 23.57 times more abundant in PH8 promastigotes according to proteomic analysis, is an interesting target due to its membrane location. Although frequently related to drug resistance, the role of ABC transporters in virulence was already reported in *Leishmania* [46–48]. Indeed, the deletion of LABCG1-2 in *Leishmania major* reduced promastigotes' infectivity *in vitro* and lesion development *in vivo*. Accordingly, LABCG1-2 deletion decreased phosphatidylserine (PS) exposure, increased parasite susceptibility to human complement lysis and decreased the proportion of metacyclics in stationary cultures, although transgenic parasites had growth curves similar to wild type counterparts [48]. These results are similar to what we found in LV79 and PH8: similar growth curves but differences in metacyclic composition.

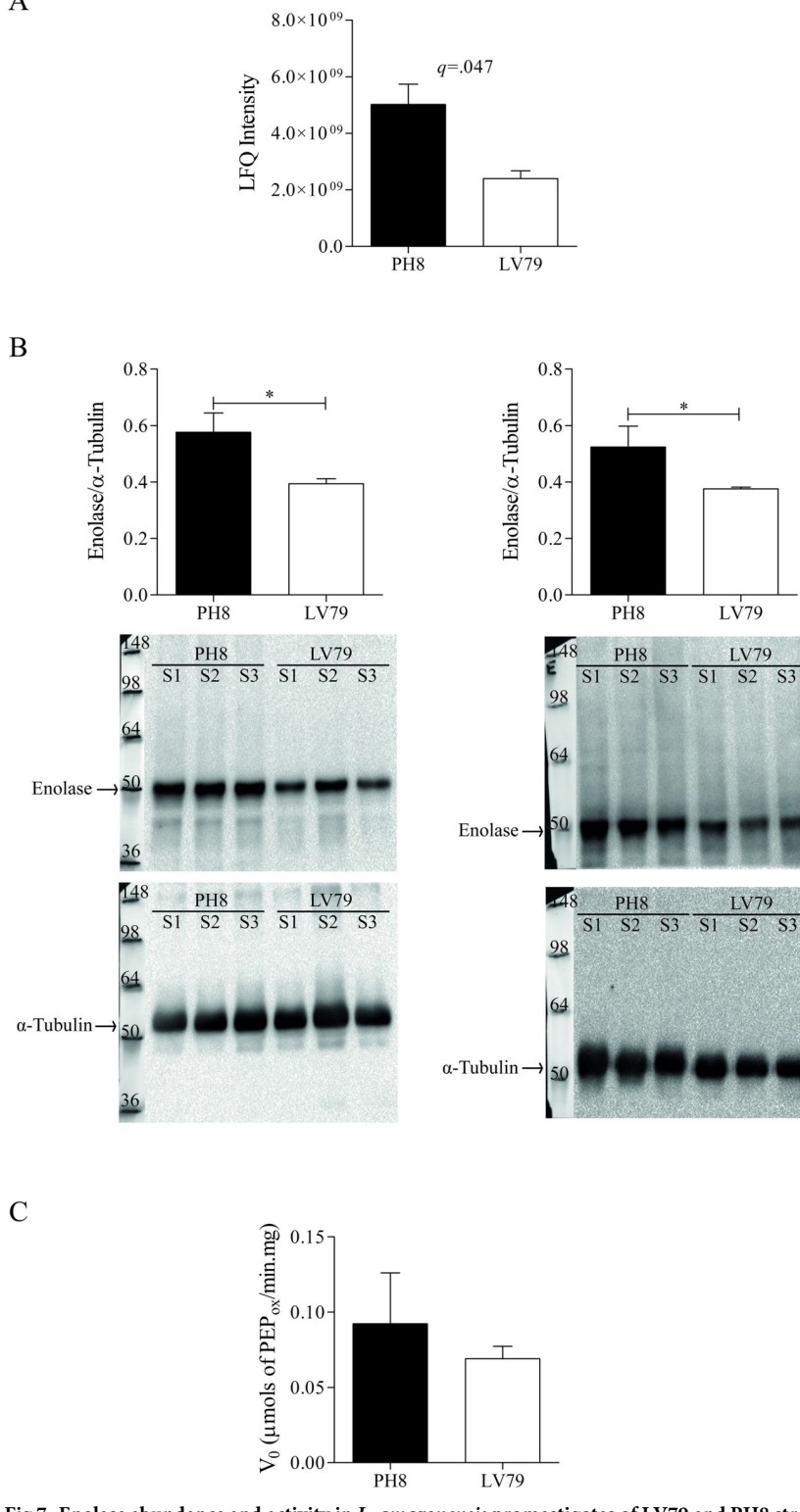

**Fig 7. Enolase abundance and activity in *L. amazonensis* promastigotes of LV79 and PH8 strains.** A. Enolase abundance (non-log transformed LFQ intensities) in proteomes of membrane-enriched fractions of LV79 and PH8. B. Enolase abundance in total extracts (left) and membrane-enriched fractions (right) (three biological samples (S1, S2 and S3) for each strain) of LV79 and PH8 by western blot. C. Enolase activity in total extracts was monitored by the PEP conversion to 2-PGA, which was measured spectrophotometrically at 240 nm. Data in A, B and C represent

means and SD of three, three and five biological replicates, respectively. For A, statistical analysis was performed by Student's *t* test with Benjamini-Hochberg correction (FDR = 0.05) and the resulting *q*-value is shown in graph. For B and C, statistical analysis was performed by Student's *t*-test, *: $p < .05$, **: $p < .01$, ***: $p < .001$.

Enolase is an enzyme that participates in glycolysis and gluconeogenesis, however an inactive form associated with plasma membrane was already demonstrated in *L. major*, *L. mexicana* and *L. donovani* [49, 50]. Enolase present on the parasite's surface may bind to mammalian host plasminogen, which can be converted into plasmin [51]. There are evidences that plasminogen is beneficial to the parasite, since plasminogen-defective male mice develop smaller lesions when infected with *L. mexicana* [52]. It is hypothesized that plasmin bound to enolase allows parasite dispersion through fibrin degradation [52]. Macrophages can also bind plasminogen through surface receptors; therefore, enolase may facilitate parasite's interaction with the host cell [53]. Furthermore, plasminogen may also inhibit complement system, conferring additional advantage to *Leishmania* [54]. Interestingly, enolase was recently described in exosomes shed by *L. amazonensis* promastigotes [16]. It is possible that the increase in

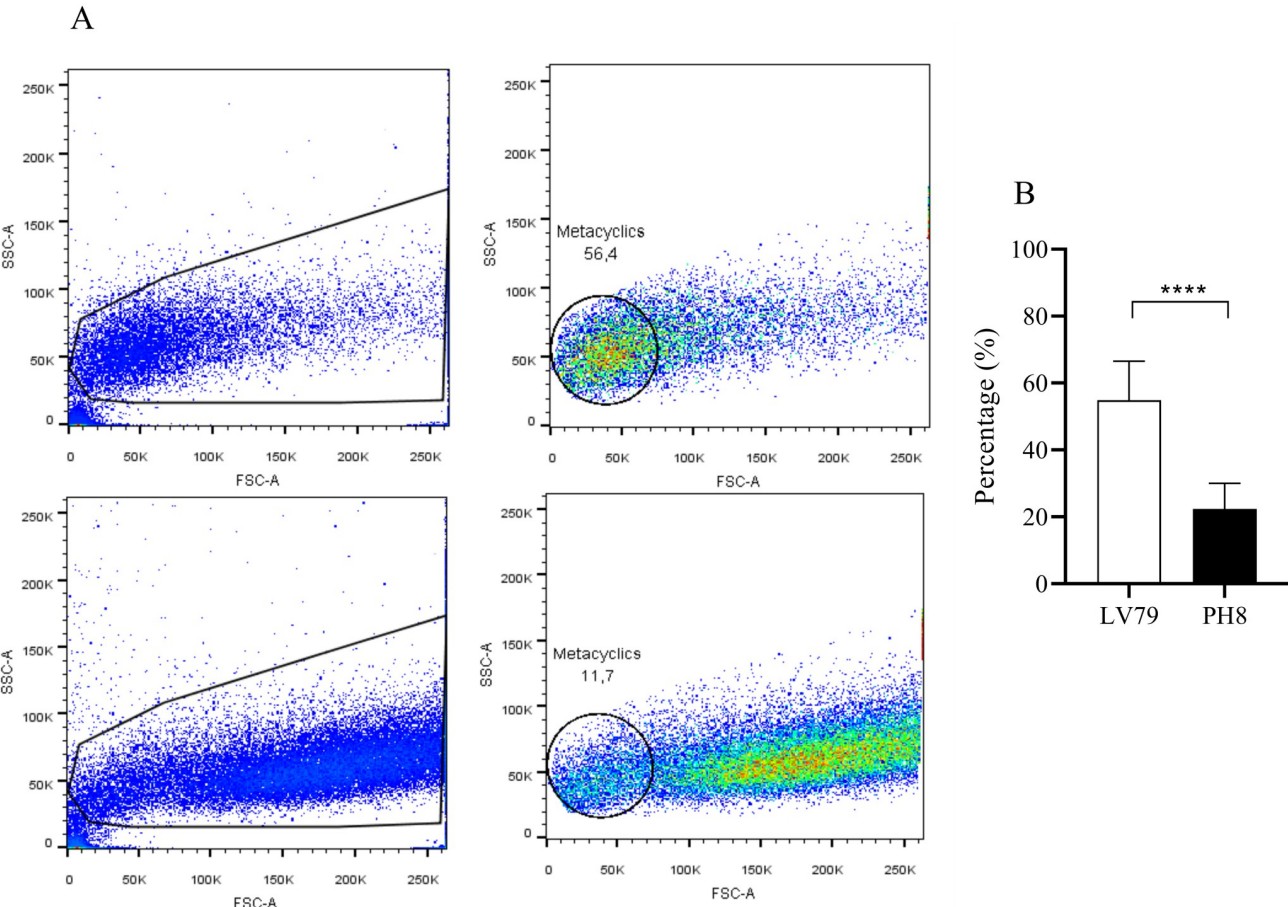

**Fig 8. Frequency of metacyclic promastigote forms in *L. amazonensis* LV79 and PH8 strains.** *L. amazonensis* promastigotes from LV79 and PH8 strains at early stationary phase (day 4) were analyzed by flow cytometry. A. Dot plots of SSC and FSC features representative of *Leishmania* gating strategy in FSC^low and SSC, excluding debris, analyzed using BD LSR Fortessa Cell Analyser (Becton Dickinson). B. Frequency of metacyclic promastigotes in three independent experiments containing 5 replicates of each strain. Each bar represents the mean of the percent ±SD of metacyclics. Statistical analysis by Student's *t*-test, ****: p < 0.0001.

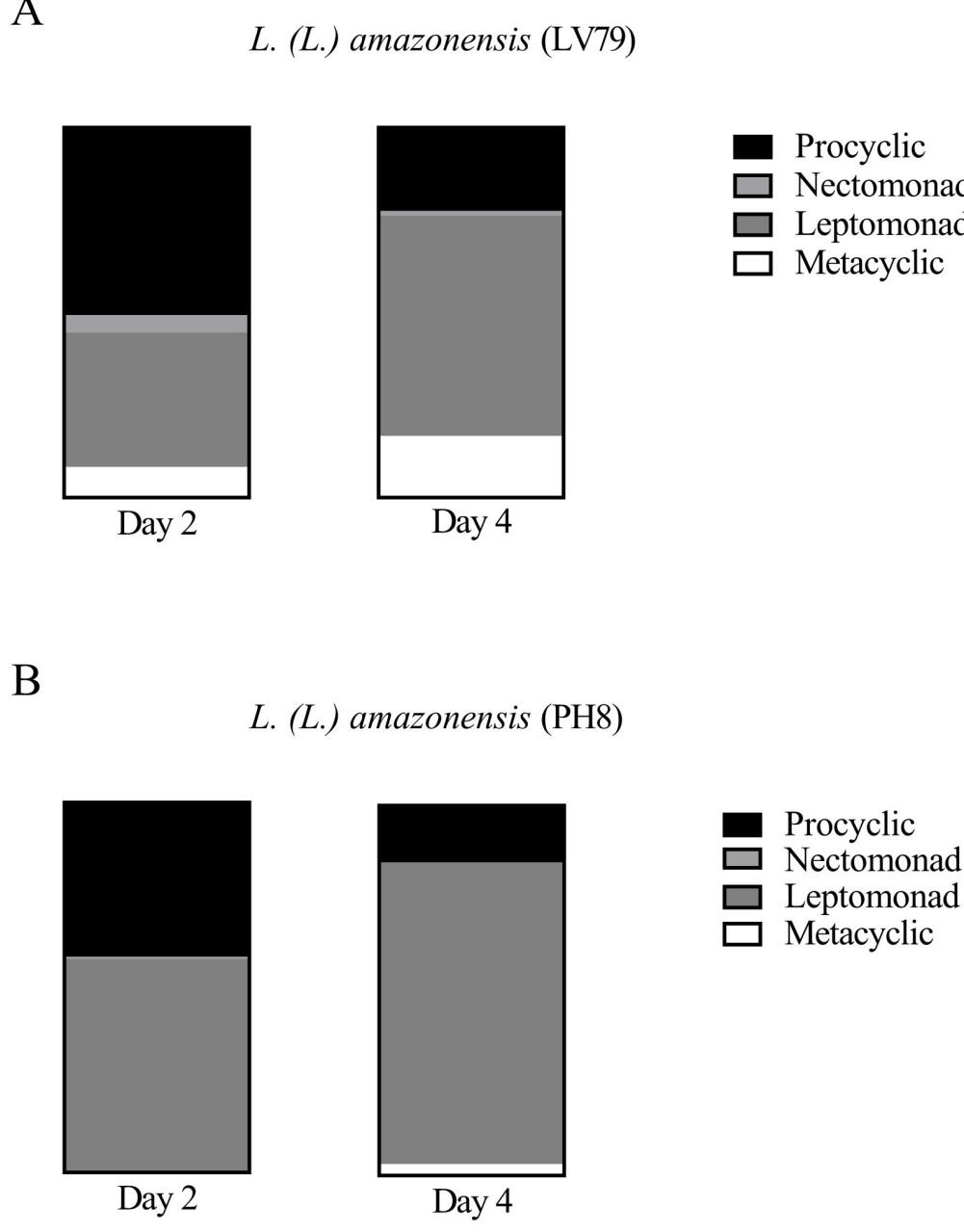

**Fig 9. Proportion of procyclics, nectomonads, leptomonads and metacyclics in log phase (day 2) and early stationary phase (day 4) cultures of *L. (L.) amazonensis* LV79 (A) and PH8 (B) strains.** Flagellum and body length from 150 parasites were measured using ImageJ and parasites were classified according to [21].

enolase observed in PH8 promastigotes contributes to the higher infectivity of this strain. Enolase identified in our proteomes may be membrane-associated or cytosolic. In fact, apart from membrane-associated proteins, we also identified several metabolic enzymes in membrane-enriched fractions of PH8 and LV79 promastigotes. These findings are curious but not unexpected, since enzymes may be closely associated to membranes of cell or organelles and may also be kept inside vesicles formed during membrane rupture. Although the protocol

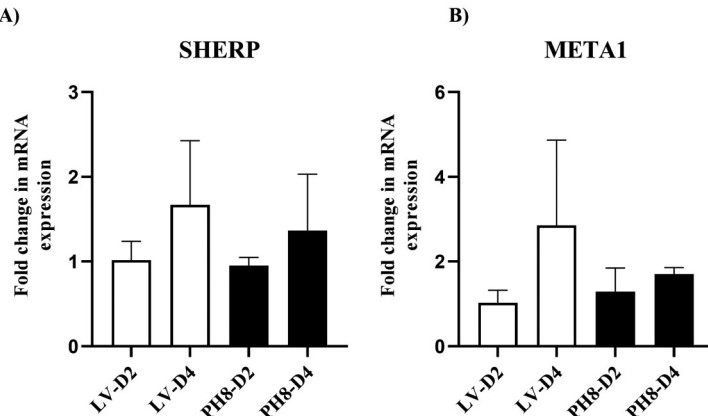

**Fig 10. Relative expression of SHERP and META1 in LV79 and PH8 cultures.** Synchronized cultures from day 2 (D2) and day 4 (D4) were collected for comparative analysis of transcripts of SHERP, META1 and GAPDH. Data obtained by RealTime RT-PCR of three independent cultures for each strain, using GAPDH as the reference gene and mean values of LV79 D2 for normalization. Statistical analysis by ANOVA with Tukey posttest.

employed sodium carbonate, which should reduce vesicle formation, these artifacts may still occur [23]. No significant difference was observed in enolase enzymatic activity between total extracts of PH8 and LV79, which may suggest that the inactive enolase isoform is responsible for differences of abundance observed between strains.

The virulence factor GP63 was also identified as differentially abundant between membrane-enriched fractions of LV79 and PH8. This glycoprotein is more abundant in promastigotes than amastigotes and is important in several phases of infection [13]. GP63 is secreted by promastigotes in exosomes, which are inoculated in the vertebrate by the sandfly during the blood meal [13, 55–57]. Besides, GP63 present in exosomes enhances cutaneous leishmaniasis in *L. amazonensis* experimental model [16]. In the extracellular matrix, it may cleave collagen and fibronectin, helping in promastigote movement [58]. It may also cleave and inactivate C3b, interrupting complement cascade but enabling parasite internalization by CR3 [59, 60]. Binding of GP63 to CR3 inhibits pro-inflammatory signaling and oxidative burst, contributing to parasite survival [59, 61, 62]. Inside the macrophage, GP63 translocates to cytoplasm and nucleus, where it affects several mammalian host cell pathways [13, 14, 17]. Curiously, according to proteomics analysis GP63 was more abundant in membrane-enriched fractions of LV79, which was confirmed by western blot. However, zymography (based on a band with the same migration of GP63 in non-reducing WB) suggests that GP63 proteolytic activity is higher in PH8 promastigotes and therefore may contribute to its virulence. These results, which must be confirmed by other assays, may indicate that GP63 activity is not directly proportional to its expression, however there is little information about the regulation of GP63 activity in *Leishmania*.

Although not directly related to phagocytosis, metabolic differences may also be important to understand the different behavior previously reported for PH8 and LV79 *in vivo*. The majority of the proteins upregulated in LV79 promastigotes participate in translation. In contrast, most of the proteins upregulated in PH8 are involved in carbohydrate metabolism, cytoskeleton composition and vesicle and membrane trafficking. Several studies have compared procyclic and metacyclic promastigotes of different *Leishmania* species [63–65]. During metacyclogenesis of *L. major*, proteins related to translation are downregulated and proteins related to motility, upregulated [64]. Furthermore, in *L. infantum* and *L. tropica* proteins that

participate of carbohydrate metabolism are more abundant in metacyclics promastigotes [63, 65]. There are no similar studies for *L. (L.) amazonensis* in the literature, but these observations prompted us to analyze whether a higher abundance of metacyclics in PH8 day 4 cultures could account for increased infectivity and different proteome profile of this strain.

The comparison of metacyclic proportions between *L. (L.) amazonensis* strains is not a simple task. Purification of *L. amazonensis* metacyclics is usually performed using antibody mAb 3A1 directed to procyclic LPG [66] or Ficoll gradients. Ficoll separation was developed for *L. major* [67], and although not based on LPG and feasible for purification of LPG deficient *L. major*, its efficiency is affected by LPG characteristics. The LPG profile of the PH8 strain revealed presence of glucose as side chains [68], and for this reason we don´t consider the comparison of the proportion of metacyclics in PH8 and LV79 by Ficoll purification accurate.

In face of these technical caveats on metacyclic purification for comparison, we opted to estimate metacyclic proportions by two different morphological techniques and to compare the abundance of commonly used metacyclic markers in cultures of LV79 and PH8. We observed higher abundance of metacyclics in LV79 according to the morphological analyses, although the expression of molecular markers was similar between the two strains. These data reinforce the complexity of classification of stages in *L. amazonensis* and defy the scientific community to invest in projects aimed to study *L. amazonensis* development in the vector and search for appropriate metacyclic markers for this species.

We believe our work contributes to different aspects of *L. amazonensis* biology. First, it compares strains with different biological behavior and identifies proteins more abundant in the more infective (and virulent) PH8 strain, some of which can be further studied as potentially involved in *Leishmania* adhesion and phagocytosis. Besides, we highlight the importance of evaluating not only the abundance of the proteins, but also their biological activity, as speculated for GP63. We also show that although PH8 parasites have a proteomic profile more similar to metacyclics, their morphological characteristics indicate a higher proportion of procyclics. These data reinforce the complexity of classification of stages in *L. (L.) amazonensis* and defy the scientific community to invest in projects aimed to study *L. (L.) amazonensis* development in the vector and search for appropriate metacyclic markers for this species. We also believe our work may stimulate further studies comparing PH8 and LV79, such as those focused on the analysis of exosome composition. Although our proteome study didn't include analysis of exosomes, we do believe PH8 and LV79 secrete exosomes with different composition in terms of membrane and cytosolic proteins (as well as other molecules). Comparison of the two secretomes will certainly add further information to *L. amazonensis* virulence and pathogenicity.

## Supporting information

**S1 Fig. Growth curve of *L. amazonensis* PH8 and LV79 promastigotes.** PH8 and LV79 promastigotes were cultured in 199 medium at 24˚C and culture density was calculated daily over 6 days. Cultures were initiated with $2 \times 10^6$ promastigotes/mL on day 0. Data represented as mean ± SD of three biological replicates.
(TIF)

**S2 Fig. Membrane enrichment of LV79 and PH8 promastigotes extracts.** Before proteomics analysis, membrane enrichment was confirmed by Western blot of cytoplasmic (C) and membrane-enriched (M) extracts with anti-GP63 and anti-TXNPx antibodies. Data shown is representative of three analyses performed with different extracts from paired PH8 and LV79 promastigotes cultures.
(TIF)

**S3 Fig. Representative images of LV79 and PH8 cultures at days 2, 4 and 6.** Cultures were synchronized, and parasites were counted, fixed and stained with Giemsa at days 2, 4 and 6 of culture.
(TIF)

**S1 Table. All proteins identified with at least two valid numbers in one of the groups (PH8 or LV79) after exclusion of contaminants, reverse hits and proteins only identified by site.** The LFQ Intensity values were transformed into log2 (X).
(XLSX)

**S2 Table. List of proteins exclusively detected in PH8 samples.** Uniprot identification (protein ID), protein names and iBAQ (sum of the intensities of the peptides normalized by the number of peptides theoretically formed) of proteins exclusively detected in PH8.
(XLSX)

**S3 Table. List of proteins exclusively detected in LV79 samples.** Uniprot identification (protein ID), protein names and iBAQ (sum of the intensities of the peptides normalized by the number of peptides theoretically formed) of proteins exclusively detected in LV79.
(XLSX)

**S4 Table. Differentially abundant proteins identified after T test with Benjamini-Hochberg correction and FDR = 0.05.** Protein names, Uniprot identification (protein ID), fold change PH8/LV79, q-value (corrected p-value) and biological process of differentially abundant proteins.
(XLSX)

**S5 Table. Protein subcellular localization.** Subcellular localization of the proteins identified in proteomic analysis was established according to Gene ontology annotations (Cellular Component). For those that lack annotation, subcellular location was determined based on a consensus between DeepLoc-1.0, WoLF PSORT and Loc Tree3 predictions. The proteins were classified as secreted or belonging to the "cilium", cytosol, golgi apparatus, glycosome, cell membrane, mitochondria, nucleus, endoplasmic reticulum (ER) or intracellular vesicles. In addition, the proteins were distinguished between soluble and integral membrane proteins according to DeepLoc-1.0 and TOPCONS predictions.
(XLSX)

**S1 Raw images.**
(PDF)

## Acknowledgments

We would like to thank Daniel Quina for help with zymography, Rob McMaster for anti-GP63 antibody, Eduardo Coelho for anti-enolase antibody and Angela Kaysel Cruz for anti-TXNPx antibody. We also thank Silvia Uliana for discussions about this work.

## Author Contributions

**Conceptualization:** Beatriz Simonsen Stolf.

**Data curation:** Fabia Tomie Tano.

**Formal analysis:** Fabia Tomie Tano, Giuseppe Palmisano.

**Funding acquisition:** Beatriz Simonsen Stolf.

**Investigation:** Fabia Tomie Tano, Gustavo Rolim Barbosa, Eloiza de Rezende, Rodolpho Ornitz Oliveira Souza, Sandra Marcia Muxel.

**Methodology:** Fabia Tomie Tano, Gustavo Rolim Barbosa, Rodolpho Ornitz Oliveira Souza, Sandra Marcia Muxel, Ariel Mariano Silber, Giuseppe Palmisano, Beatriz Simonsen Stolf.

**Project administration:** Fabia Tomie Tano, Beatriz Simonsen Stolf.

**Resources:** Beatriz Simonsen Stolf.

**Supervision:** Beatriz Simonsen Stolf.

**Validation:** Fabia Tomie Tano, Ariel Mariano Silber, Giuseppe Palmisano, Beatriz Simonsen Stolf.

**Visualization:** Fabia Tomie Tano.

**Writing – original draft:** Fabia Tomie Tano, Beatriz Simonsen Stolf.

**Writing – review & editing:** Fabia Tomie Tano, Rodolpho Ornitz Oliveira Souza, Ariel Mariano Silber, Giuseppe Palmisano, Beatriz Simonsen Stolf.

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
