## [Decision Letter · Decision Letter 0]

12 Apr 2022

PONE-D-22-07087Proteome and morphological analysis show unexpected differences between promastigotes of L. amazonensis PH8 and LV79 strainsPLOS ONE

Dear Dr. Stolf,

Thank you for submitting your manuscript to PLOS ONE. After careful consideration, we feel that it has merit but does not fully meet PLOS ONE’s publication criteria as it currently stands. Therefore, we invite you to submit a revised version of the manuscript that addresses the points raised during the review process.

We look forward to receiving your revised manuscript.

Kind regards,

Yara M. Traub-Csekö

Academic Editor

PLOS ONE

Journal Requirements:

Reviewers' comments:

Reviewer's Responses to Questions

**Comments to the Author**

1. Is the manuscript technically sound, and do the data support the conclusions?

Reviewer #1: Yes

Reviewer #2: Yes

Reviewer #3: Partly

2. Has the statistical analysis been performed appropriately and rigorously? 

Reviewer #1: Yes

Reviewer #2: Yes

Reviewer #3: Yes

3. Have the authors made all data underlying the findings in their manuscript fully available?

Reviewer #1: Yes

Reviewer #2: Yes

Reviewer #3: Yes

4. Is the manuscript presented in an intelligible fashion and written in standard English?

Reviewer #1: Yes

Reviewer #2: Yes

Reviewer #3: Yes

5. Review Comments to the Author

Reviewer #1: In the manuscript entitled "Proteome and morphological analysis show unexpected differences between promastigotes of L. amazonensis PH8 and LV79 strains” has been used two strains of Leishmania amazonensis (PH8 and LV79), causing different clinical presentations of cutaneous leishmaniases in mice, as shown in their previous publication (reference 15). In present study, were compared promastigotes’ infectivity in macrophages, their proteomes and morphologies. The authors try to correlate the biological findings with the identified proteins. Apart from comparing protein abundance, they also analyzed enzymatic activity when appropriate, and noticed that protein abundance do not always correspond to activity.

In general, the study is well written and experiments have been conducted with appropriate controls, replication, and sample sizes. The conclusions drawn appropriately based on the data presented, perhaps except for the activity of GP63.

In my opinion, it does not seem possible to assume that the zymography data can be attributed exclusively to GP63 activity. Hence I would like to know “ how” and “why” the authors assume that “protein abundance do not always correspond to activity” performing only zymography.

It seems to me that the authors have already answered several questions from other reviewers, as well as improved writing and data presentation, and now the article is almost suitable for publication.

- Minor Revision (Line numbers are missing, which can make it difficult to find the text)

ABSTRACT

I cannot understand how “contradictory results shed light”…. it looks confused to me

INTRODUCTION

Page 4- The authors use “vector and host”, however, vector is also host

M & M

2.3 - What is the quantity and/or volume of parasites? 1X not required. PBS pH?

2.5 – Number of cells are missing in some protocols. Fixed with 4% paraformaldehyde (Water or PBS?)

2.12- Primer sequence of HASP (or HASPA)? After all, was HASP (or HASPA?) used or not? It is not described here, but it appears in the result (page 25), and in the caption of Figure 10, but not in the bars of Figure 10!

Reviewer #2: The work presented by the authors aims to analyze the protein composition of two strains of Leishmania amazonensis (PH8 and LV79) that show differences in relation to the infectivity of murine macrophages. In this work, a proteomic analysis of material enriched with membrane proteins was performed. The authors performed in vitro adhesion and phagocytosis assays which revealed that the PH8 strain has a greater adhesion capacity and is also more endocytosed by macrophages than the LV79 strain. Validation experiments of data obtained in proteomics were performed using techniques such as Western blot and Real time RT-PCR, which provided robustness to the analyses. The result obtained with the GP63 zymogram could be more accurate and revealing if the authors had used a fluorescent peptide substrate, such as Dansyl-AYKKWV-NH2. Overall the manuscript is well written in correct and objective English.

Reviewer #3: The manuscript PONE-D-22-07087 reports on the in vitro infection capability and proteomics analysis of two Leishmania amazonensis strains associated with different severity of cutaneous leishmaniasis in BALB/c mice. Bone marrow-derived macrophages were used for comparing parasites’ adherence and infectivity and membrane-enriched fractions of early stationary phase promastigotes were analyzed by label-free proteomics. The study is technically fair; however, there are some issues that authors are encouraged to address.

Major comments:

- Pag. 6. The end of the introduction is a summary of the results. Instead of this summary, the authors could clarify in that final paragraph what was the hypothesis and the objective of the study, since it is not clear why to return to an in vitro infection with promastigotes after having made a comparative analysis of amastigotes obtained from in vivo infections. On the one hand the authors mention phenotypic differences (emphasizing membrane proteins), on the other they seem to be interested in metacyclogenesis. So, it is not clear in the text, what is the main objective.

- Pag. 10. Authors state: “For total protein extracts (used in SDS-PAGE, western blot and zymography), promastigotes were resuspended in PBS with protease inhibitor cocktail (Fermentas) at a final density of 2 x 109 parasites/mL. Parasites were lysed by eight cycles of freeze and thaw (liquid nitrogen and 40oC).” This way of lysing parasites favors soluble proteins while membrane proteins remain in the debris. Please clarify which fraction was used for the SDS-PAGE, WB and Zym assays?

- Pag. 16. Item 3.1. During the encounter of the parasites with the phagocytic cells, both the parasites actively infect the cells (when they are infectiuos) and the cells actively phagocytize the parasites. Decrease in the infective capability of the former or phagocytic capability of the latter would result in lower rates of infection. It is not clear to this reviewer why the authors describe the results in terms of phagocytosis and make no mention of the infectivity of the strains. It is also not clear why, if they talk about phagocytosis, there is no specific control for this, such as latex beads.

- Figure 1. Figure 1A, describing the results in terms of 500 cells is not clear. Please describe the number of parasites attached per cell (in the population of 500 cells tested) or alternatively the percentage of cells with parasites attached (in the population of 500 cells tested). The same applies to figure 1B, describe the number of phagocytosed parasites per cell.

- Pag. 18. Figure 2. As described in the methods, the total protein extracts for SDS-PAGE and Western blotting were prepared differently than the membrane and cytoplasmic fractions. In figure 2, total protein only appears to represent the soluble fraction, so it would be more correct to present the total protein extract from which the M and C fractions were prepared.

- Page 18. Supplementary table 1. For quantification and clusterization, proteins identified with only one peptide should be excluded. Please, exclude those proteins.

- Page 19. Supplementary table 4. This table presents fold change values, but it is not clear if these are based on the LFQ or iBAQ values. Please clarify that in the legend of the table.

Regarding the iBAQ values, could it be clarified in the text why these values were included and how they were used in the analysis of the results?

- Pag. 20. The authors report that only 14% of the identified proteome has at least one transmembrane domain, which reveals that proteins from other cell compartments would be the majority in the analyzed membrane fraction. Given this finding, the data should be better filtered based on the results of the predictors so that only proteins with predicted transmembrane domains are included in the final list, giving more robustness to the result. For this reviewer, 14% (~230 proteins), being stringent and restricted to only true membrane proteins, would be a much more interesting group to analyze and discuss. Alternatively, an analysis of the cytosolic fraction could be done for comparison and subtraction from the membrane fraction.

- Page 21. First paragraph. Once again, these results show that the data probably needs to be better filtered so as not to lose focus on the membrane proteins involved in adhesion and infection.

- Pag 21. Figure 6. How do the authors explain that the total extracts of both strains do not contain what was observed in the membrane fractions? Please explain.

- Page 22. First paragraph and figure 6C. It is not clear why the authors credit the 50 kDa band to GP63. In zymography the electrophoretic migration of proteins is retarded, so enzyme activity may appear in regions above (not below) the calculated theoretical molecular mass for the protein. Thus, the molecular mass values of the observed proteolytic activities should be reviewed in figure 6C. Moreover, assigning a proteolytic band to GP63 solely based on electrophoretic migration is wrong. The most that can be said is that they are metallopeptidases that migrate in that molecular mass and even so, this figure lacks a control with a specific inhibitor for this class of enzymes, to demonstrate that they are metallopeptidases and not another class of peptidases. That control should be included.

Also, please describe in the legend to figure 6 what panel C-right is.

- Page 23. Item 3.7, first paragraph. For this reviewer, this is due to the limitations of the methodology used for the enrichment of membrane proteins and it is again suggested to better filter the data so that proteins without transmembrane domains are not included.

- Fig. 8. Cytometry profiles are very different between strains on day 4 of culture. While in the growth curve, on day 4 the strains seem to have the same number of parasites and to be in the same growth phase (early stationary), the analysis by cytometry shows completely different profiles. How do the authors explain that?

- Pag. 26-27. Authors state: “Only 3% of the proteins were localized in the cell membrane, probably because sodium carbonate extraction promotes the enrichment of integral membrane proteins, including those belonging to intracellular organelles. Although the percentage of integral membrane proteins is low, other studies that perform cell fractionation of Leishmania promastigotes by differential centrifugation have also failed to achieve a significant enrichment [44, 45].” If authors filter better the data, I suggest correcting this percentage for the 14% reported before (proteins with transmembrane proteins). I also suggest rewriting this paragraph since it does not seem correct to justify the low number of membrane proteins obtained in this study, using the failure that other studies have shown in this enrichment.

- Pag. 27. Interestingly, ABCG1 is a phosphatidylserine transporter potentially involved in oxidative stress and metacyclogenesis among others (Parasites Vectors 10, 267 (2017). https://doi.org/10.1186/s13071-017-2198-1). The authors are encouraged to elaborate on this.

- Pag. 28. The authors cannot claim that the observed proteolytic activities are due to GP63. Please re-write that paragraph.

- Page 29. This reviewer suggests avoiding discussing metabolic differences as these proteins appear to be "contaminants" of the membrane fractions and there are no other assays showing differences in the mentioned metabolic pathways between the strains.

Minor comments:

• Page 3 Introduction. Whenever the subgenus is mentioned, the genus should be mentioned first. Please correct for L. Leishmania and L. Viannia

• Pag. 4. Leishmania infantum chagasi is not a species, L. chagasi is a synonym of L. infantum.

• Pag. 4. It is important to mention that the main characteristic of the diffuse form caused by L. amazonensis is the absence of cell-mediated immunity.

• Pag. 6. Authors state: “… This conflicting data indicates that metacyclogenesis in L. amazonensis is a complex issue and that different methods must be used to characterize parasite stages and to search for factors involved in infectivity.” The results shown in the manuscript do not support this statement. Please moderate to sentence.

• Pag. 7. Item 2.2. This reviewer suggests making a more complete description of these strains, such as their origin and the host from which they were isolated.

• Page 8. Item 2.4. Please, clearly describe how the synchronization of the cultures was done.

• Page 8. Last paragraph. Please explain why this infection was made at that temperature (4°C) and how it reflects (or not) the natural interaction of these cells (macrophages-parasites)

• Supplementary figure 2. Instead of being supplemental, this figure could be added to the current figure 2.

• In figure 10 the titles of the Y axis are missing

6. PLOS authors have the option to publish the peer review history of their article (what does this mean?). If published, this will include your full peer review and any attached files.

Reviewer #1: No

Reviewer #2: No

Reviewer #3: No

---

## [Author Response · Author response to Decision Letter 0]

6 Jun 2022

Dear Dr. Yara Traub-Csekö, 

We want to thank you and the reviewers for the opportunity of submitting a revised version of our work. We also want to thank the three reviewers for the critical and detailed analysis that contributed for a better version of the manuscript. We believe and hope that this new version of the paper is suitable for publication in PlosOne. 

Please find our answers to reviewers´requests below:

Reviewer #1: In the manuscript entitled "Proteome and morphological analysis show unexpected differences between promastigotes of L. amazonensis PH8 and LV79 strains” has been used two strains of Leishmania amazonensis (PH8 and LV79), causing different clinical presentations of cutaneous leishmaniases in mice, as shown in their previous publication (reference 15). In present study, were compared promastigotes’ infectivity in macrophages, their proteomes and morphologies. The authors try to correlate the biological findings with the identified proteins. Apart from comparing protein abundance, they also analyzed enzymatic activity when appropriate, and noticed that protein abundance do not always correspond to activity.

In general, the study is well written and experiments have been conducted with appropriate controls, replication, and sample sizes. The conclusions drawn appropriately based on the data presented, perhaps except for the activity of GP63.

R: We would like to thank reviewer 1 for the positive analysis of our work in terms of the conduction of experiments and text quality.

In my opinion, it does not seem possible to assume that the zymography data can be attributed exclusively to GP63 activity. Hence I would like to know “ how” and “why” the authors assume that “protein abundance do not always correspond to activity” performing only zymography.

R: We have performed a Western blot of GP63 under non-reducing conditions to map GP63 position in conditions similar to those used in zymography. As seen in Fig6C (right panel), GP63 band in non-reducing gel migrates near 50KDa. Zymography shown in Fig6C (left panel) shows a band near 50KDa at the same position of GP63 in the corresponding WB. Besides, zymography was done using samples containing several protein inhibitors, to guarantee that metalloprotease activity would be preserved, but may other proteases would be inactive.

Other works have used similar approaches, such as the ones by Duque et al., 2019 and Hassani et al., 2014.

It seems to me that the authors have already answered several questions from other reviewers, as well as improved writing and data presentation, and now the article is almost suitable for publication.

- Minor Revision (Line numbers are missing, which can make it difficult to find the text)

R: We apologize for that. Line numbers were included.

ABSTRACT

I cannot understand how “contradictory results shed light”…. it looks confused to me

R: We agree with this reviewer. We have modified the text to “These results shed light to the gaps in our knowledge of metacyclogenesis in L. amazonensis and to proteins that should be studied in the context of infection by this species.”

INTRODUCTION

Page 4- The authors use “vector and host”, however, vector is also host

R: The reviewer is correct. We have changed host for mammalian host. 

M & M

2.3 - What is the quantity and/or volume of parasites? 1X not required. PBS pH?

R: 2 x 106 promastigotes were collected in each of the 5 tubes prepared for each strain. This information was added to the text. Thank you.

2.5 – Number of cells are missing in some protocols. Fixed with 4% paraformaldehyde (Water or PBS?)

R: The number of cells was mentioned in the text: “Cells stained with 0.4% trypan blue were counted using a hemocytometer and 4 x 105 viable cells were plated… macrophages were incubated with parasites at multiplicity of infection (MOI) of 10:1”. Cells were fixed in PBS containing 4% paraformaldehyde. 

2.12- Primer sequence of HASP (or HASPA)? After all, was HASP (or HASPA?) used or not? It is not described here, but it appears in the result (page 25), and in the caption of Figure 10, but not in the bars of Figure 10!

R: Sorry, results obtained for HASP were not of good quality, and thus were not included in the figure. I apologize for not removing from the previous version of the text.

Reviewer #2: The work presented by the authors aims to analyze the protein composition of two strains of Leishmania amazonensis (PH8 and LV79) that show differences in relation to the infectivity of murine macrophages. In this work, a proteomic analysis of material enriched with membrane proteins was performed. The authors performed in vitro adhesion and phagocytosis assays which revealed that the PH8 strain has a greater adhesion capacity and is also more endocytosed by macrophages than the LV79 strain. Validation experiments of data obtained in proteomics were performed using techniques such as Western blot and Real time RT-PCR, which provided robustness to the analyses. The result obtained with the GP63 zymogram could be more accurate and revealing if the authors had used a fluorescent peptide substrate, such as Dansyl-AYKKWV-NH2. Overall the manuscript is well written in correct and objective English.

R: We would like to thank reviewer 2 for the positive analysis of our work in terms of experiments, validations and text quality. We agree that the use of fluorescent peptides in activity assays would add more precision to GP63 analysis. Unfortunately, we are not able to do this assay at this moment, but we will include it in future experiments. 

Reviewer #3: The manuscript PONE-D-22-07087 reports on the in vitro infection capability and proteomics analysis of two Leishmania amazonensis strains associated with different severity of cutaneous leishmaniasis in BALB/c mice. Bone marrow-derived macrophages were used for comparing parasites’ adherence and infectivity and membrane-enriched fractions of early stationary phase promastigotes were analyzed by label-free proteomics. The study is technically fair; however, there are some issues that authors are encouraged to address.

R: We would like to thank this reviewer for careful revision and for the precise summary of our work.

Major comments:

- Pag. 6. The end of the introduction is a summary of the results. Instead of this summary, the authors could clarify in that final paragraph what was the hypothesis and the objective of the study, since it is not clear why to return to an in vitro infection with promastigotes after having made a comparative analysis of amastigotes obtained from in vivo infections. On the one hand the authors mention phenotypic differences (emphasizing membrane proteins), on the other they seem to be interested in metacyclogenesis. So, it is not clear in the text, what is the main objective.

R: Thank you for the suggestion. We have modified this last paragraph, removing repetition and explaining the aim of the study.

- Pag. 10. Authors state: “For total protein extracts (used in SDS-PAGE, western blot and zymography), promastigotes were resuspended in PBS with protease inhibitor cocktail (Fermentas) at a final density of 2 x 109 parasites/mL. Parasites were lysed by eight cycles of freeze and thaw (liquid nitrogen and 40oC).” This way of lysing parasites favors soluble proteins while membrane proteins remain in the debris. Please clarify which fraction was used for the SDS-PAGE, WB and Zym assays?

R: For total extracts we lysed promastigotes by freeze and thaw and used the lysate without prior centrifugation. Thus, both soluble and membrane proteins were present.

- Pag. 16. Item 3.1. During the encounter of the parasites with the phagocytic cells, both the parasites actively infect the cells (when they are infectiuos) and the cells actively phagocytize the parasites. Decrease in the infective capability of the former or phagocytic capability of the latter would result in lower rates of infection. It is not clear to this reviewer why the authors describe the results in terms of phagocytosis and make no mention of the infectivity of the strains. It is also not clear why, if they talk about phagocytosis, there is no specific control for this, such as latex beads.

R: The reviewer has raised an interesting “contradiction” on the use of the term infectivity. It is considered that Leishmania does not actively infect or invade macrophages; instead, they are phagocytosed. However, the term infective capacity or infectivity is frequently employed. 

The same macrophage was employed for both strains, so the difference in binding and phagocytosis is due to differences in surface molecules of the two strains. Since we wanted to have a comparative (qualitative) analysis of binding and internalization of PH8 and LV79, we didn´t employ latex beads.

- Figure 1. Figure 1A, describing the results in terms of 500 cells is not clear. Please describe the number of parasites attached per cell (in the population of 500 cells tested) or alternatively the percentage of cells with parasites attached (in the population of 500 cells tested). The same applies to figure 1B, describe the number of phagocytosed parasites per cell.

R: We opted to mention parasites attached in 500 cells because the numbers of parasites were a little lower than usual in this experiment. In fact, due to the short time of parasite: macrophage contact (5min) in phagocytosis assay, numbers of parasites attached and phagocytosed are sometimes quite low. We do agree that mentioning parasites per cell would be better, but we would have to show decimal numbers. Since the aim of the experiment is a comparative view of the two strains, we chose to represent in 500 cells, showing numbers above 1.

- Pag. 18. Figure 2. As described in the methods, the total protein extracts for SDS-PAGE and Western blotting were prepared differently than the membrane and cytoplasmic fractions. In figure 2, total protein only appears to represent the soluble fraction, so it would be more correct to present the total protein extract from which the M and C fractions were prepared.

R: The reviewer has raised an interesting point about the way we prepared the total protein mentioned in figure 2. We could have done this extract in parallel and with the same solutions we used for membrane and cytoplasmic fractions obtained by ultracentrifugation, so that the comparison would be more precise. Anyway, we consider that membrane x cytoplasmic comparison is the most important for validation of the extraction protocol, so we believe figure 2 is valid.

- Page 18. Supplementary table 1. For quantification and clusterization, proteins identified with only one peptide should be excluded. Please, exclude those proteins.

R: For protein identification, at least one razor + unique peptide was required. We chose this criterium because several Leishmania proteins, such as GP63, belong to multigene families, and a combination of razor and unique peptides allows for a confident identification of proteins within these families. For quantification, we required at least two ratio counts. Sup table 1 shows all proteins identified and quantified using the parameters described above. 

- Page 19. Supplementary table 4. This table presents fold change values, but it is not clear if these are based on the LFQ or iBAQ values. Please clarify that in the legend of the table.

R: Fold changes are based on LFQ. This information is mentioned in the legend: “fold change PH8/LV79 (based on LFQ Intensity values)”.

Regarding the iBAQ values, could it be clarified in the text why these values were included and how they were used in the analysis of the results?

R: iBAQ values were included as an additional parameter but were not employed for other calculations. iBAQ was requested by a former reviewer, and we opted to keep it only in the table.

LFQ (label-free quantitation) values were used for the quantitative comparison between the two strains and downstream analyses.

- Pag. 20. The authors report that only 14% of the identified proteome has at least one transmembrane domain, which reveals that proteins from other cell compartments would be the majority in the analyzed membrane fraction. Given this finding, the data should be better filtered based on the results of the predictors so that only proteins with predicted transmembrane domains are included in the final list, giving more robustness to the result. For this reviewer, 14% (~230 proteins), being stringent and restricted to only true membrane proteins, would be a much more interesting group to analyze and discuss. Alternatively, an analysis of the cytosolic fraction could be done for comparison and subtraction from the membrane fraction.

R: The prediction of transmembrane (TM) domains is based on the in-silico translated amino acid sequences obtained by genomic data. However, in Leishmania (and other trypanosomatids) there are several membrane proteins that do not have TM domains but that are anchored by GPI anchors and other structures added after posttranslational modifications. Besides, there are proteins that are membrane associated and have roles outside the cell. 

In order to identify differentially modified proteins and map the majority of regulated proteins, we initially considered the entire dataset without filtering proteins with TM domains. 

Furthermore, we validated a set of selected proteins focusing mainly on membrane proteins.

- Page 21. First paragraph. Once again, these results show that the data probably needs to be better filtered so as not to lose focus on the membrane proteins involved in adhesion and infection.

R: We believe the reviewer refers to the biological process related to the regulated proteins. Indeed, differentially regulated proteins are mostly associated to translation, carbohydrate, amino acid or nucleotide metabolism, cytoskeleton composition and vesicle and membrane trafficking, functions not related to adhesion and infection. As mentioned in the previous question, we opted not to filter TM proteins because many Leishmania proteins can be located in the plasma membrane even without TM domains. Besides, classification in processes can be sometimes misleading because proteins can have more than one function. Indeed, processes such as “proteolysis” and “carbohydrate metabolism” can include proteins known to be also located in membrane, as GP63 and enolase, respectively. Additionally, many proteins are not classified in processes, although certainly located in the membrane. One example is Putative ATP-binding cassette protein subfamily G, member 1 (E9AKN6), 23 times more abundant in PH8 (sup table IV).

- Pag 21. Figure 6. How do the authors explain that the total extracts of both strains do not contain what was observed in the membrane fractions? Please explain.

R: I am not sure if we understood the reviewer´s comment. GP63 was quantified in Western blot of total and membrane-enriched fractions. In WB of membrane-enriched fractions, GP63 labeling is shown together with tubulin labeling, so we observe more bands. We have included this information in the figure legend.

- Page 22. First paragraph and figure 6C. It is not clear why the authors credit the 50 kDa band to GP63. In zymography the electrophoretic migration of proteins is retarded, so enzyme activity may appear in regions above (not below) the calculated theoretical molecular mass for the protein. Thus, the molecular mass values of the observed proteolytic activities should be reviewed in figure 6C. Moreover, assigning a proteolytic band to GP63 solely based on electrophoretic migration is wrong. The most that can be said is that they are metallopeptidases that migrate in that molecular mass and even so, this figure lacks a control with a specific inhibitor for this class of enzymes, to demonstrate that they are metallopeptidases and not another class of peptidases. That control should be included.

Also, please describe in the legend to figure 6 what panel C-right is.

R: The reviewer is correct to expect that migration should be retarded in zymography. However, the non-reducing Western blot for GP63 shown in the right panel of Fig 6C indicates an accelerated migration for GP63 in such conditions. Bouvier et al., 1985 reported similar findings in a previous study (see reference at the end of the document).

We agree with the reviewer that we cannot precisely assign the band to GP63 without the counterproof with a specific inhibitor. We have modified the text accordingly. Panel C-right is a non-reducing Western blot for GP63, included as reference of migration. Sorry for not mentioning it. This information was included in the legend.

- Page 23. Item 3.7, first paragraph. For this reviewer, this is due to the limitations of the methodology used for the enrichment of membrane proteins and it is again suggested to better filter the data so that proteins without transmembrane domains are not included.

R: As mentioned on a previous answer, we agree that several proteins contribute to these “phenotypes”, but we opted not to filter data in order not to lose potential membrane proteins. Since all extracts were processed in parallel and using the same protocol, we compared samples in terms of all proteins identified and proteins of interest were validated by different methods.

- Fig. 8. Cytometry profiles are very different between strains on day 4 of culture. While in the growth curve, on day 4 the strains seem to have the same number of parasites and to be in the same growth phase (early stationary), the analysis by cytometry shows completely different profiles. How do the authors explain that?

R: We expected that LV79 and PH8 would have similar culture compositions on day 4, since growth curves were very similar. Cytometry experiments were performed to clarify the differences observed in the proteomes, which suggested that PH8 cultures had more metacyclic promastigotes than LV79. Curiously, cytometry profiles suggested that LV79 had more metacyclics than PH8, an observation supported by morphometric analysis. Although we have no explanation for these observations, differences in metacyclic composition were already reported in parasite cultures with similar growth curves. Indeed, Manzano et al showed that the deletion of LABCG genes 1 and 2 in L. major increased parasite susceptibility to human complement lysis and decreased the proportion of metacyclics in stationary cultures, although transgenic parasites had growth curves similar to wild type counterparts. These results are similar to what we found in LV79 and PH8: similar growth curves but differences in metacyclic composition. 

- Pag. 26-27. Authors state: “Only 3% of the proteins were localized in the cell membrane, probably because sodium carbonate extraction promotes the enrichment of integral membrane proteins, including those belonging to intracellular organelles. Although the percentage of integral membrane proteins is low, other studies that perform cell fractionation of Leishmania promastigotes by differential centrifugation have also failed to achieve a significant enrichment [44, 45].” If authors filter better the data, I suggest correcting this percentage for the 14% reported before (proteins with transmembrane proteins). I also suggest rewriting this paragraph since it does not seem correct to justify the low number of membrane proteins obtained in this study, using the failure that other studies have shown in this enrichment.

R: We thank the reviewer for the suggestions. As we mentioned previously, cell membrane proteins may have TMs, may have non-protein anchors or may be bound to other proteins. Thus, we cannot replace this number by TM proteins. We opted to remove the information about proteins in cell membrane from the text, and not to mention failures from other studies, as suggested by the reviewer. Thank you.

- Pag. 27. Interestingly, ABCG1 is a phosphatidylserine transporter potentially involved in oxidative stress and metacyclogenesis among others (Parasites Vectors 10, 267 (2017). https://doi.org/10.1186/s13071-017-2198-1). The authors are encouraged to elaborate on this.

R: The reviewer has called attention to a very interesting subject. Indeed, the mentioned paper shows that LABCG1 and 2 have PS floppase activity, are involved in autophagy and redox metabolism and affect infectivity, virulence and metacyclogenesis. PH8 strain, which shows much higher levels of LABCG1 than LV79, is also more infective and virulent. We have included this information in the text. Thank you.

- Pag. 28. The authors cannot claim that the observed proteolytic activities are due to GP63. Please re-write that paragraph.

R: We do agree that our results suggest that GP63 activity is higher in PH8, but other assays are required to prove that. We have modified the text accordingly: “However, zymography (based on a band with the same migration of GP63 in non-reducing WB) suggests that GP63 proteolytic activity is higher in PH8 promastigotes and therefore may contribute to its virulence These results, which must be confirmed by other assays, may indicate that GP63 activity is not directly proportional to its expression, however there is little information about the regulation of GP63 activity in Leishmania.

- Page 29. This reviewer suggests avoiding discussing metabolic differences as these proteins appear to be "contaminants" of the membrane fractions and there are no other assays showing differences in the mentioned metabolic pathways between the strains.

R: Although our focus was on membrane proteins, the low enrichment of these proteins called attention to unexpected differences in other categories. The distinct profile of the two strains in terms of protein categories related to metabolic process suggests that the differences are robust and real. We opted to include discussion about metabolic differences since it is known that some Leishmania enzymes may also be located on the surface, sometimes with “moonlight functions” such as enolase, and to stimulate other groups to study LV79 and PH8 and other Leishmania strains in terms of metabolic differences.

Minor comments:

• Page 3 Introduction. Whenever the subgenus is mentioned, the genus should be mentioned first. Please correct for L. Leishmania and L. Viannia

R: We apologize for the mistake. Thank you very much for the careful revision.

• Pag. 4. Leishmania infantum chagasi is not a species, L. chagasi is a synonym of L. infantum.

R: We agree and revised the text. 

• Pag. 4. It is important to mention that the main characteristic of the diffuse form caused by L. amazonensis is the absence of cell-mediated immunity.

R: Thank you. We have added this information.

• Pag. 6. Authors state: “… This conflicting data indicates that metacyclogenesis in L. amazonensis is a complex issue and that different methods must be used to characterize parasite stages and to search for factors involved in infectivity.” The results shown in the manuscript do not support this statement. Please moderate to sentence.

R: We have removed this phrase from page 6 and have focused on the main aims of the paper. Thanks for the suggestion.

• Pag. 7. Item 2.2. This reviewer suggests making a more complete description of these strains, such as their origin and the host from which they were isolated.

R: We have added your suggestion to the text, thank you. L. amazonensis PH8 strain (IFLA/ BR/1967/PH8) was isolated from the sand fly Lutzomyia flaviscutellata from Pará State, Brazil, while L. amazonensis LV79 (MPRO/BR/72/M 1841) was obtained from the rodent Proechimys sp also from Pará.

• Page 8. Item 2.4. Please, clearly describe how the synchronization of the cultures was done.

R: Sorry for the incomplete description. We have modified it to “Cultures were synchronized by sub culturing 3 times for 2x106/mL every 3 days”.

• Page 8. Last paragraph. Please explain why this infection was made at that temperature (4°C) and how it reflects (or not) the natural interaction of these cells (macrophages-parasites)

R: The incubation at 4oC for 2 hours allows parasite contact with macrophage but not internalization. For a “adhesion” or “binding” assay, cells would be fixed after this step. The further incubation at 34oC 5% CO2 for 5 minutes used in our protocol allows phagocytosis of Leishmania. We have employed the same protocol in Galuppo et al. 

• Supplementary figure 2. Instead of being supplemental, this figure could be added to the current figure 2.

R: We opted to keep it as supplemental since validation of the membrane- enrichment protocol was performed before this experiment and is shown in figure 2. 

• In figure 10 the titles of the Y axis are missing

R: We apologize for this mistake. Y axis should be “fold change in mRNA expression”

REFERENCES 

Guillermo Arango Duque, Armando Jardim, Étienne Gagnon, Mitsunori Fukuda, Albert Descoteaux.The host cell secretory pathway mediates the export of Leishmania virulence factors out of the parasitophorous vacuole. PLoS Pathog. 2019 Jul 29;15(7): e1007982. doi: 10.1371/journal.ppat.1007982. 

Kasra Hassani, Marina Tiemi Shio, Caroline Martel, Denis Faubert, Martin Olivier. Absence of metalloprotease GP63 alters the protein content of Leishmania exosomes. PLoS One .2014 Apr 15;9(4):e95007. doi: 10.1371/journal.pone.0095007.

Bouvier J.; Etges, R. J.; Bordier, C. Identification and purification of membrane and soluble forms of the major surface protein of Leishmania promastigotes. Journal of Biological Chemistry, v. 260, n. 29, p. 15504-15509, dez. 1985

Manzano JI, Perea A, León-Guerrero D, Campos-Salinas J, Piacenza L, Castanys S, Gamarro F. Leishmania LABCG1 and LABCG2 transporters are involved in virulence and oxidative stress: functional linkage with autophagy. Parasit Vectors. 2017 May 30;10(1):267. doi: 10.1186/s13071-017-2198-1.

Galuppo MK, de Rezende E, Forti FL, Cortez M, Cruz MC, Teixeira AA, Giordano RJ, Stolf BS. CD100/Sema4D Increases Macrophage Infection by Leishmania (Leishmania) amazonensis in a CD72 Dependent Manner. Front Microbiol. 2018 Jun 5;9:1177. doi: 10.3389/fmicb.2018.01177

---

## [Decision Letter · Decision Letter 1]

5 Jul 2022

Proteome and morphological analysis show unexpected differences between promastigotes of Leishmania amazonensis PH8 and LV79 strains

PONE-D-22-07087R1

Dear Dr. Stolf

We’re pleased to inform you that your manuscript has been judged scientifically suitable for publication and will be formally accepted for publication once it meets all outstanding technical requirements.

Kind regards,

Yara M. Traub-Csekö

Academic Editor

PLOS ONE

Additional Editor Comments (optional):

Reviewers' comments:

Reviewer's Responses to Questions

**Comments to the Author**

1. If the authors have adequately addressed your comments raised in a previous round of review and you feel that this manuscript is now acceptable for publication, you may indicate that here to bypass the “Comments to the Author” section, enter your conflict of interest statement in the “Confidential to Editor” section, and submit your "Accept" recommendation.

Reviewer #2: All comments have been addressed

Reviewer #3: All comments have been addressed

2. Is the manuscript technically sound, and do the data support the conclusions?

Reviewer #2: Yes

Reviewer #3: Partly

3. Has the statistical analysis been performed appropriately and rigorously? 

Reviewer #2: Yes

Reviewer #3: I Don't Know

4. Have the authors made all data underlying the findings in their manuscript fully available?

Reviewer #2: Yes

Reviewer #3: Yes

5. Is the manuscript presented in an intelligible fashion and written in standard English?

Reviewer #2: Yes

Reviewer #3: Yes

6. Review Comments to the Author

Reviewer #2: The authors answered all my questions and I consider the article ready to be published. So I have nothing more to add.

Reviewer #3: (No Response)

7. PLOS authors have the option to publish the peer review history of their article (what does this mean?). If published, this will include your full peer review and any attached files.

Reviewer #2: No

Reviewer #3: No

---

## [Editor Report · Acceptance letter]

11 Aug 2022

PONE-D-22-07087R1 

Proteome and morphological analysis show unexpected
differences between promastigotes of *Leishmania
amazonensis* PH8 and LV79 strains 

Dear Dr. Stolf:

I'm pleased to inform you that your manuscript has been deemed suitable for publication in PLOS ONE. Congratulations! Your manuscript is now with our production department. 

Kind regards, 

on behalf of

Dr. Yara M. Traub-Csekö 

Academic Editor

PLOS ONE